# DNA methylation and gene expression changes derived from assisted reproductive technologies can be decreased by reproductive fluids

Sebastian Canovas[1,2], Elena Ivanova[3], Raquel Romar[1,2], Soledad García-Martínez[1,2], Cristina Soriano-Úbeda[1,2], Francisco A García-Vázquez[1,2], Heba Saadeh[3,4†], Simon Andrews[4], Gavin Kelsey[3,5], Pilar Coy[1,2]*

[1]Physiology of Reproduction Group, Departamento de Fisiología, Facultad de Veterinaria, Universidad de Murcia-Campus Mare Nostrum, Murcia, Spain; [2]Instituto Murciano de Investigación Biosanitaria, Murcia, Spain; [3]Epigenetics Programme, The Babraham Institute, Cambridge, United Kingdom; [4]Bioinformatics Group, The Babraham Institute, Cambridge, United Kingdom; [5]Centre for Trophoblast Research, University of Cambridge, Cambridge, United Kingdom

**Abstract** The number of children born since the origin of Assisted Reproductive Technologies (ART) exceeds 5 million. The majority seem healthy, but a higher frequency of defects has been reported among ART-conceived infants, suggesting an epigenetic cost. We report the first whole-genome DNA methylation datasets from single pig blastocysts showing differences between in vivo and in vitro produced embryos. Blastocysts were produced in vitro either without (C-IVF) or in the presence of natural reproductive fluids (Natur-IVF). Natur-IVF embryos were of higher quality than C-IVF in terms of cell number and hatching ability. RNA-Seq and DNA methylation analyses showed that Natur-IVF embryos have expression and methylation patterns closer to in vivo blastocysts. Genes involved in reprogramming, imprinting and development were affected by culture, with fewer aberrations in Natur-IVF embryos. Methylation analysis detected methylated changes in C-IVF, but not in Natur-IVF, at genes whose methylation could be critical, such as IGF2R and NNAT.

*For correspondence: pcoy@um.es

Present address: †Computer Science Department,KASIT, The University of Jordan, Amman, Jordan

Competing interests: The authors declare that no competing interests exist.

## Introduction

"Most fertility researchers are trying to improve Assisted Reproductive Technologies (ARTs) success as measured by a single, clear standard: the birth of an apparently healthy baby. Only a few are trying to discern whether in vitro fertilization (IVF) leaves a subtle legacy in children. What will happen to these kids when they are middle-aged?" (*Servick, 2014*). In humans, according to a study by the World Health Organisation (WHO) in 190 countries, infertility affects 20% of couples and it was estimated that at least 40.5 million women were seeking infertility medical care in 2007 (*Mascarenhas et al., 2012*). ARTs provide a helpful alternative for a high proportion of infertility cases and the number of children born to date using these methods exceeds 5 million (*International Committee for Monitoring Assisted Reproductive Technology I, 2012*). Although the majority of them seem healthy, studies have reported higher rates of preterm births (*Rubens et al., 2014*), non-chromosomal birth defects and adverse perinatal effects in ART pregnancies (*El Hajj and Haaf, 2013*), with long-term effects being under study in humans (*Kissin et al.,*

**eLife digest** Infertility has become more common in many countries, particularly those where many people delay having children until later in life. To help individuals experiencing infertility conceive a child, scientists have developed treatments called assisted reproductive technologies (or ARTs for short). So far, more than 5 million children have been born with the help of these treatments. Most of the children seem healthy; however, birth defects are more common in ART-conceived babies than those conceived without treatment.

The cause of these birth defects is not known, though scientists suspect it may have something to do with techniques used in ART. One possible culprit is the liquid that is used in the laboratory to help the parents' sperm and egg come together for fertilization. This same liquid is also used to bathe the developing embryo for the first few days after fertilization before it is implanted into its mother's womb. Some scientists wonder whether adding the fluids normally found in the reproductive tract of their mother to this liquid could reduce defects in children conceived via ART.

Now, Canovas et al. have shown that fertilizing and growing pig embryos in liquids supplemented with fluid from the wombs of female pigs results in embryos that are closer to naturally conceived pig embryos than in non-supplemented liquids. In the experiments, naturally conceived embryos were compared to ART embryos exposed to the usual liquids and with ART embryos grown in liquids with fluid collected from the pig's reproductive tract added. Cutting edge technologies were used to sequence the entire genomes of all of the embryos and compare which genes were active in each case. Canovas et al. also looked at chemical markers on the DNA – called epigenetic changes – that turn on or off the expression of genes without changing the DNA code itself.

The analysis showed that ART-conceived embryos grown in the usual liquid had different patterns of gene expression and epigenetic changes compared to naturally conceived embryos. Gene expression and epigenetic changes in the ART embryos grown with the pig reproductive fluid was more similar to the naturally conceived embryos.

These findings suggest that abnormal gene expression in the ART-liquid exposed embryos may lead to birth defects, and that using natural reproductive fluids may be safer. To confirm this, scientists will have to implant embryos conceived in these three different conditions into mother pigs and assess the health and gene expression patterns of the resulting piglets. If successful, these new insights might one day lead to improvements in ART techniques used to treat infertility in people.

*2014*). Epidemiological data suggest that perturbed epigenetic gene regulation by the application of ART could be a contributory factor in these adverse outcomes (*El Hajj and Haaf, 2013*; *Whitelaw et al., 2014*), although such alterations could also be considered as consequences of parental characteristics, gamete quality or other non-epigenetic technique-derived effects (*Simpson, 2014*). To clarify the impact of each of these factors, the use of an animal model that avoids, as much as possible, the effect of parental circumstances and the use of protocols minimizing the technique-derived effects would help to attain the goal of offering safer ART for patients.

For modeling ART-related disorders in human, swine could be a good candidate for several reasons: their genetic, anatomical and physiological similarities with human (*Swindle et al., 2012*); their size and length of gestation; and the availability of individuals genetically selected by their excellent reproductive performance in artificial insemination centres. Importantly, this last trait could be useful to remove the paternal factor (low-quality male gametes) from studies as a possible reason for any epigenetic alterations found. However, most protocols for processing boar spermatozoa for IVF include their selection by density gradient centrifugations and just a few used the swim-up procedure to isolate highly motile spermatozoa which is the routine selection in human infertility clinics. Since it was observed that spermatozoa selected by swim-up show higher rates of normal morphology and motility, and decreased DNA fragmentation and methylation levels (*Kim et al., 2015*), it would be necessary to adapt the sperm selection protocols in pig before using them to model ART-derived epigenetic alterations.

In both mouse and human, accumulating evidence indicates that the embryo is sensitive to its very early environment and that culture media used in ART (as factors involved in technique-derived effects) may have long-lasting consequences (*Kleijkers et al., 2014*; *Fernandez-Gonzalez et al., 2004*). Several imprinting disorders and abnormal phenotypes have been linked to ART, but of special significance is the relationship between the presence of serum in culture media and the incidence of Large Offspring syndrome (LOS) in ruminants (*Young et al., 1998*), which includes diverse pathologic alterations and shows phenotypic and epigenetic similarities with the imprinted disorder Beckwith-Wiedemann syndrome (BWS) in humans (*Chen et al., 2013*). Since it was proposed that serum in the culture medium could be a crucial factor in LOS incidence, the tendency in the procedures for both human and livestock was to move toward the use of chemically defined media, limiting the presence of proteins in the culture medium to serum albumin. Although practical, this approach may have unpredictable consequences, because it ignores the fact that the reproductive fluids have a different composition to serum and are extremely rich in proteins other than serum albumin (more than 150 have been described in the oviductal fluid [*Avilés et al., 2010*]). If these proteins are physiologically present, they must play a variety of roles supporting the normal development of the embryo, roles that serum albumin alone cannot properly provide and serum cannot fully mimic. In addition, although ART in species such as cattle and sheep usually results in foetal overgrowth (*Young et al., 2001*; *Chen et al., 2015*), opposing phenotypes such as low birth weights (excluding BWS) are often seen in humans (*Schieve et al., 2002*) and pigs (*García-Vázquez et al., 2010*). A study showing the relationship between child birth weight and the protein source in embryo culture media (*Zhu et al., 2014*) reinforces the hypothesis that the protein composition of the culture media plays a role in the correct regulation of epigenetic marks in the growing embryo. A similar conclusion can be reached from a clinical trial showing that protein enrichment of media compared with addition of serum albumin alone improved the blastocyst implantation rate and may increase human births by more than 8% (*Meintjes et al., 2009*). Therefore, as with breast milk, which is so complex and so rich in bioactive factors that cannot be easily replaced with any artificial composition (*Hennet and Borsig, 2016*), the idea that reproductive secretions could be necessary in the culture media should not be underrated. At least, it should be explored under experimental conditions to unveil the relevance of these secretions.

DNA and RNA sequencing have become affordable cutting-edge technologies that could help to understand the mechanisms underlying abnormalities observed in ART-derived offspring. However, so far, single blastocyst whole-genome DNA methylation profiles comparing in vivo and in vitro produced embryos have not been published for any mammalian species and we therefore aim to produce these in this study.

We report here that modified swim-up protocols for the selection of spermatozoa in pigs and the use of reproductive secretions as additives in the culture media significantly increase the yield and quality of the blastocysts produced from a morphological, epigenetic and gene expression point of view. Using genome-wide analyses of gene expression by RNA-Seq and DNA methylation by Bisulfite-Seq in single blastocysts, we provide datasets of pig blastocysts produced in vitro with and without reproductive secretions as additives in the culture medium and show that the former are more similar to the in vivo specimens than the later. This suggests an alternative approach for conceiving healthier ART-derived children.

## Results

### Swim-up method improves the yield of pig IVF

In order to select spermatozoa before IVF, a swim-up protocol was set up and compared with a conventional selection system by density gradient centrifugations. To do this, it was necessary to design a suitable washing and sperm selection medium imitating, as far as possible, in vivo conditions (NaturARTs PIG sperm washing medium and NaturARTs PIG sperm swim-up medium, EmbryoCloud, Murcia, Spain). The swim-up medium was supplemented either with bovine serum albumin (BSA) (Swim-up BSA group) or porcine oviductal fluid (POF, Swim-up fluid group) collected at the late follicular (LF) phase of the estrous cycle (NaturARTs POF-LF, EmbryoCloud, Murcia, Spain) (*Figure 1*). All the fluids used in this study were directly aspirated from the lumen of ovarian follicles, oviducts

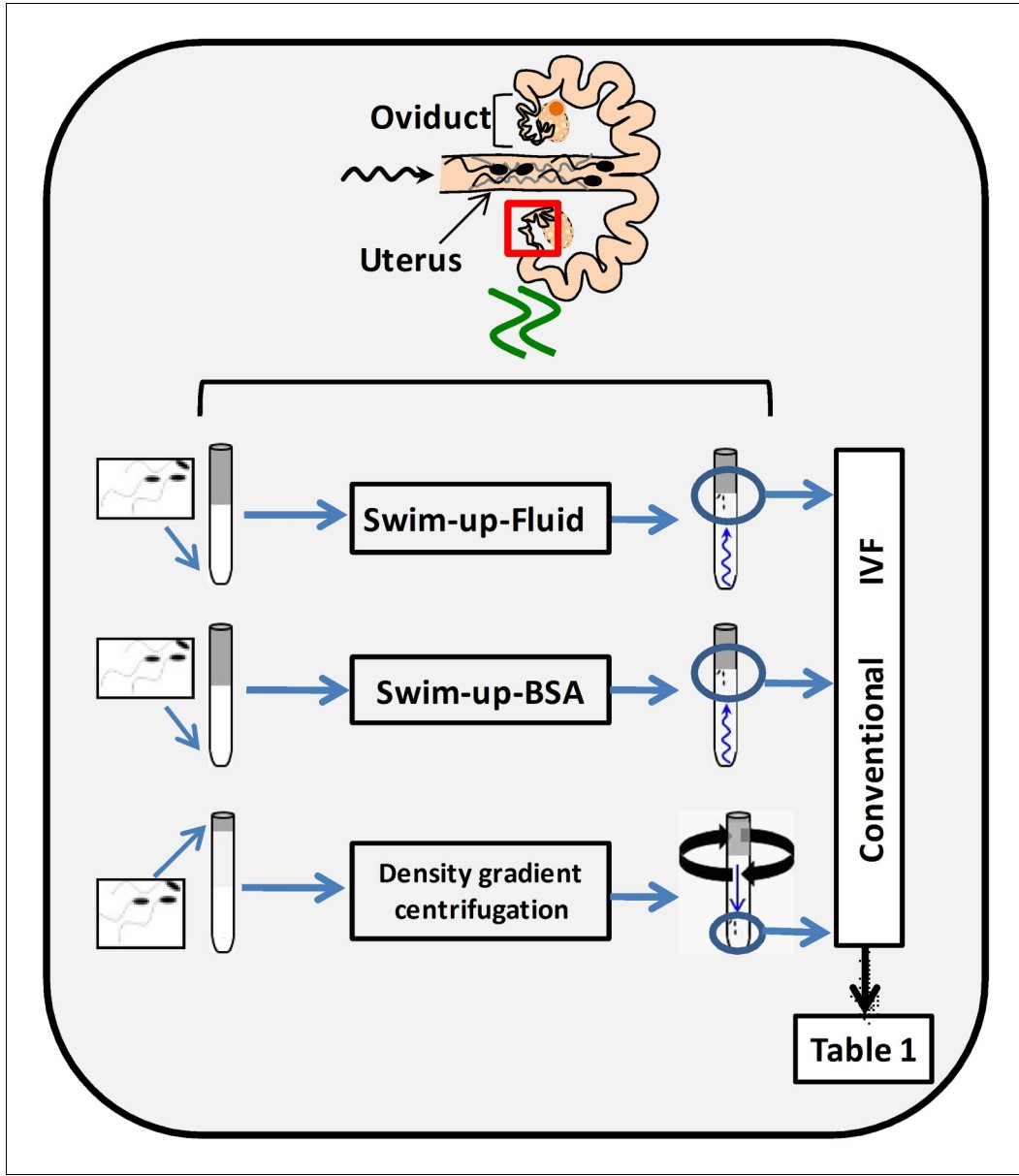

**Figure 1.** Schematic representation of three different sperm processing protocols used for in vitro fertilization. Swim-up-BSA: NaturARTs PIG medium + BSA; Swim-up-Fluid: NaturARTs PIG medium + POF-LF*. Density gradient centrifugation: centrifugation through a discontinuous Percoll: gradient (45% and 90% v/v). *POF-LF: porcine oviductal fluid collected at the late follicular phase of the estrous cycle. Red box represents the portion of the reproductive tract whose conditions we tried to resemble in vitro. IVF results after using these three different sperm processing protocols are included in **Table 1**.

or uterus and processed according to the information described in the Materials and methods section, at http://embryocloud.com, and in previous references (**Coy et al., 2008**).

Polyspermy after IVF is a major issue in the pig (**Coy and Avilés, 2010**). With these new protocols, we obtained significantly higher rates of monospermy than with conventional ones (49.6 ± 4.5 vs 17.4 ± 4.1, **Table 1**) and the final percentage of putative zygotes (evaluated at 24 hours post insemination, hpi) was significantly higher (35.2 ± 0.2 vs 14.6 ± 0.1, **Table 1**). Moreover, the addition of POF-LF to the Swim-up media instead of BSA increased the final yield of the system (35.2 ± 0.2 vs 29.7 ± 0.2, **Table 1**).

**Table 1.** IVF results after using three different sperm processing protocols (Density gradient, Swim-up-BSA and Swim-up-Fluid) as represented in *Figure 1*. [a,b]: Different letters in the same column indicate values statistically different (p<0.05). Penetration: proportion of oocytes penetrated by one or more spermatozoa. Monospermy: Monospermy percentage, calculated from penetrated oocytes, represents the proportion of penetrated oocytes with only one spermatozoon inside the ooplasm. Spermatozoa/Oocyte: Mean number of sperm per penetrated oocyte. Spermatozoa/ZP: Mean number of spermatozoa attached to ZP per oocyte. Yield: Percentage of putative zygotes per oocyte.

| Sperm processing method | N | Penetration (%) | Monospermy (%) | Spermatozoa/Oocyte | Spermatozoa/ZP | Zygote yield (%) |
|---|---|---|---|---|---|---|
| Density gradient centrifugation | 105 | 84.3 ± 3.6a | 17.4 ± 4.1a | 8.4 ± 0.7a | 17.3 ± 2.3a | 14.6 ± 0.1a |
| Swim-up-BSA | 180 | 69.6 ± 3.5b | 42.7 ± 4.6ab | 2.1 ± 0.1b | 7.2 ± 0.5b | 29.7 ± 0.2b |
| Swim-up-Fluid | 183 | 71.1 ± 3.4b | 49.6 ± 4.5b | 2.7 ± 0.1b | 8.6 ± 0.5b | 35.2 ± 0.2c |

## Reproductive fluids added to the culture media increase blastocyst quality

In a second experiment, and using the Swim-up protocol for sperm selection, a new IVF/Embryo culture (EC) system (Natur-IVF) was developed, which included preincubation of oocytes in oviductal fluid (NaturARTs PIG OF-LF) and the presence of reproductive fluids as additives in the IVF and EC media (0–8 hr: NaturARTs POF-LF; 8–48 hr: oviductal fluid from the early luteal–EL- phase of the estrous cycle, NaturARTs POF-EL; 48–180 hr: uterine fluid -UF-from this same phase, NaturARTs

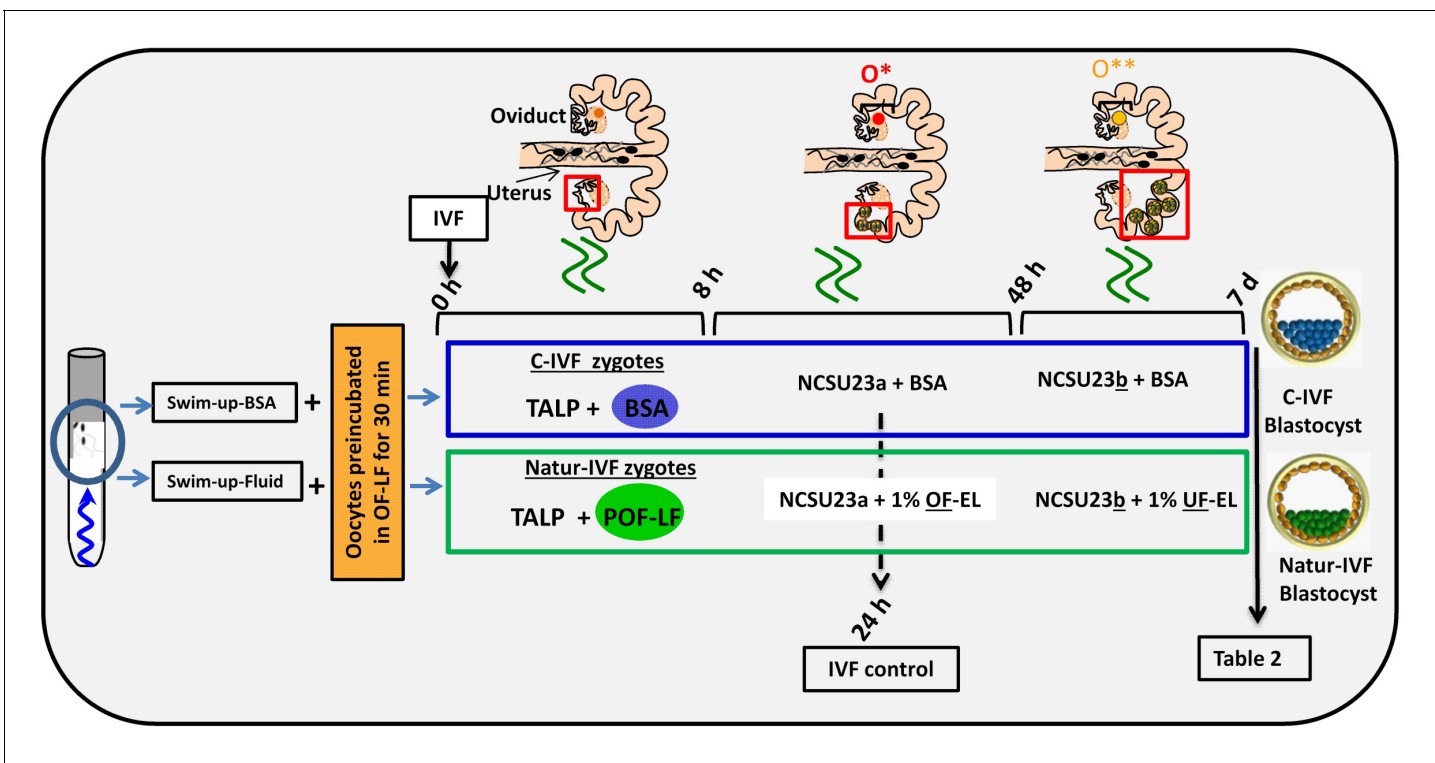

**Figure 2.** Schematic representation of the different steps of the new IVF/EC system. Swim-up-BSA or Swim-up-Fluid protocols were used for IVF. Previously, oocytes were preincubated in OF-LF for 30 min. Then, each group of putative zygotes were incubated in different media (0–8 hr, 8–48 hr and 48 hr-7days) as indicated in the diagram. O*: ovary with hemorrhagic corpus luteum; O**: early corpus luteum; OF-LF: oviductal fluid-late follicular phase of the estrous cycle; OF-EL: oviductal fluid-early luteal phase of the estrous cycle; UF-EL: uterine fluid-early luteal phase of the estrous cycle. Swim-up-BSA: NaturARTs PIG medium + BSA; Swim-up-Fluid: NaturARTs PIG medium + POF-LF. TALP: culture medium used for IVF. NCSU23: culture medium used for embryo development in vitro supplemented with sodium lactate, pyruvate and non-essential amino acids (NCSU23a) or with glucose and essential and non-essential amino acids (NCSU23b).

PUF-EL) (*Figure 2*). Corresponding controls with BSA instead of OF/UF for each step (referred as C-IVF group) were analyzed (*Figure 2*). Evaluation at 24 hpi revealed higher penetration rate (66.6 ± 0.1 *vs* 43.7 ± 0.1, p<0.05) and similar monospermy rate (78.6 ± 0.1 *vs* 72.7 ± 0.1, p<0.05) for the Natur-IVF and C-IVF groups, respectively. Regarding embryo development, more than 40% of cleaved embryos reached the blastocyst stage in both groups (*Table 2A*). However, the Natur-IVF blastocysts showed a significant increase in their mean number of cells (81.8 ± 7.2, *Table 2A*) compared to the C-IVF ones (49.9 ± 3.7), and this number was similar to that observed in the in vivo samples (In-vivo group, 87.0 ± 7.2). Moreover, at day 7.5, embryos reaching the hatching or hatched stages were observed only in the Natur-IVF group (*Table 2B*). Taken together, these data indicate a higher quality, in terms of cell number and ability to hatch, in the ART-derived blastocysts when reproductive fluids were added to the culture medium.

## The blastocyst transcriptome can be modulated in vitro by reproductive fluids

In vitro culture systems significantly alter embryonic gene expression as previously observed in pooled pig blastocysts (*Bauer et al., 2010*). Here, the transcriptomes from three individual day 7.5 blastocysts produced by C-IVF or Natur-IVF were compared with their in vivo counterparts (*Figure 3A–B*). RNA libraries showed acceptable quality in all nine blastocysts. Mean number of raw reads was 14.24 ± 2.23 (±SD) millions, and transcripts from 13,309 to 14,512 different genes (from a total of 20,789 annotated pig mRNAs) were detected in each individual. Principal Component Analysis (PCA) showed that, despite expected individual variability, the three embryos from each group clustered together (*Figure 3B*), with the C-IVF embryos showing higher variability, which could represent high embryo plasticity in response to suboptimal culture conditions. Therefore, after combining the triplicates, data from both in vitro groups showed high correlation (Pearson correlation coefficient [r] = 0.964), but Natur-IVF was closer to the In vivo group ([r] = 0.95) than C-IVF

**Table 2.** (A) Comparative results of IVF yield by using BSA (C- IVF) or reproductive fluids (Natur-IVF) as additives in the culture medium for 7.5 days. (B) Results of blastocyst development (for each type) using BSA (C- IVF) or reproductive fluids (Natur-IVF) as additives in the culture medium for 7.5 days. Columns from 'Early blastocyst' to 'Hatched blastocyst' indicate the percentage of each type of blastocyst from Total blastocyst (**Table 2A**), classified according to Bo and Mapletoft[25. a,b]: Different letters in the same column indicate values statistically different (p<0.05). Cleavage: Cleavage percentage from N. Total Blastocysts: Percentage of blastocysts calculated from cleaved embryos. Yield: Percentage of putative blastocysts from N. Cell/blastocyst: mean number of cells per blastocyst.

A)

| Group | N | Penetration (%) | Monospermy (%) | Cleavage (%) | Total blastocysts (%) | Blastocyst Yield (%) | Cell/blastocyst |
|---|---|---|---|---|---|---|---|
| In vivo | 41 | | | | | | 87.0 ± 7.2b |
| C- IVF | 903 | 395 (43.7 ± 0.1a) | 656 (72.7 ± 0.1) | 429 (47.5 ± 1.6a) | 178 (41.4 ± 2.4) | 19.6 ± 1.3 | 49.9 ± 3.7a |
| Natur-IVF | 961 | 640 (66.6 ± 0.1b) | 755 (78.6 ± 0.1) | 405 (42.1 ± 1.6b) | 180 (44.5 ± 2.5) | 18.7 ± 1.2 | 81.8 ± 7.2b |

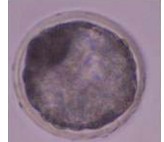 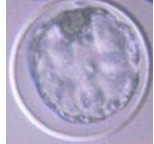 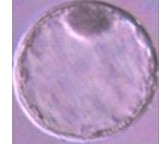 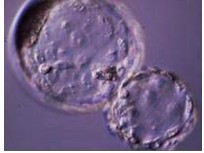 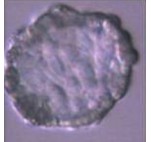

B)

| Group | N | Early blastocyst (%) | Blastocyts (%) | Expanded blastocyst (%) | Hatching blastocyst (%) | Hatched blastocyst (%) |
|---|---|---|---|---|---|---|
| C- IVF | 178 | 57 (31.7 ± 6.1)a | 50 (28.3 ± 5.9) | 71 (40.0 ± 6.4) | 0 (0)a | 0 (0)a |
| Natur -IVF | 180 | 23 (12.8 ± 5.4)b | 55 (30.8 ± 7.5) | 65 (35.9 ± 7.8) | 28 (15.4 ± 5.9)b | 9 (5.1 ± 3.6)b |

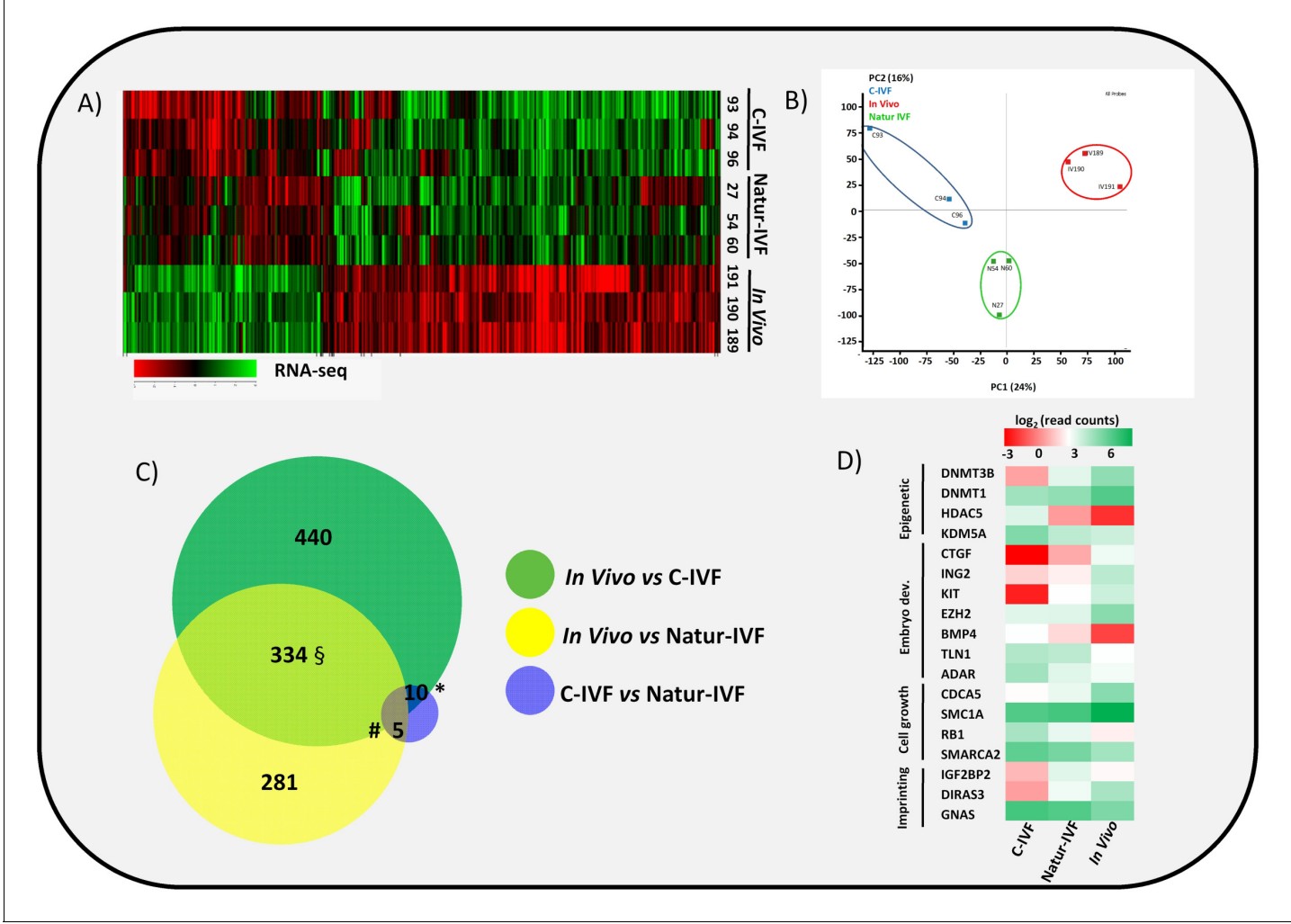

**Figure 3.** Gene expressed analysis in blastocysts obtained in vivo, by the Natur-IVF system or by C-IVF system. (**A**) Heatmap of global gene expression (with log2 fold change >1.5 and adjusted B-H p-value < 0.05). Numbers denote ID of a specific embryo. (**B**) Principal Component Analysis (PCA) of the RNA-Seq samples: In vivo embryos (IV, red), Natur-IVF (N, green) and C-IVF (C, blue). Numbers denote ID of specific embryos. (**C**) Venn diagram with DEGs (*Figure 3—source data 1*). *, #, § denotes DEGs exclusive for C-IVF, Natur-IVF and In vivo, respectively (*Figure 3—source data 2*). (**D**) Heat map of gene expression of key genes associated with embryo development/differentiation, epigenetic reprogramming, cell cycle/cell growth, gene expression and imprinting.

The following source data is available for figure 3:

**Source data 1.** Differentially expressed genes (DEGs) for pair-wise comparisons (C-IVF vs Natur-IVF, In vivovs Natur-IVF, C-IVF vs In vivo) and list of all gene expression values.

**Source data 2.** DEGs exclusives for each group: 328 DEG exclusive In vivo, 7 DEGs exclusive Natur-IVF and 13 DEGs exclusive C-IVF.

([r] = 0.938). RNA-Seq data analysis (DESeq2 p<0.05 after multiple testing correction) identified 787 differentially expressed genes (DEG) between the C-IVF and In-vivo, and 621 DEGs between Natur-IVF and In vivo (*Figure 3—source data 1*, including also all the expression values for all the genes). Of the genes that were significantly different (adjusted p-value < 0.05, fold change > 1.5) in the pair-wise comparisons, there was a higher number of up-regulated (534/787–68%- in C-IVF embryos and 431/621–69%- in Natur-IVF) than down-regulated (253 and 190, respectively) (*Figure 3C*, *Figure 3—source data 1*).

Top Canonical Pathways, Physiological Systems and Molecular and Cellular Functions related to DEGs were identified (summarized in *Supplementary file 1*) using the Ingenuity Pathway Analysis (IPA) software. Globally, down-regulated genes in C-IVF and in Natur-IVF were linked to similar Top-cellular functions (*Supplementary file 1*). Equally, top Canonical Pathways affected by up-regulated genes were similar for both groups. In contrast, two pathways were identified in down-regulated DEGs in C-IVF embryos, but not in Natur-IVF DEGs (*Supplementary file 1*). Increased pathways in Natur-IVF and C-IVF included cholesterol, mevalonate, serine and glycine biosynthesis and p53 signaling. Decreased pathways (protein ubiquitination and 14-3-3 mediated signaling) were detected only in C-IVF. Similarly, Physiological Systems and Functions over-represented by up-regulated or down-regulated DEGs were different between C-IVF or Natur-IVF. These results show that, in spite of similarity, there were differences that could influence specific pathways and affect key molecular and cellular functions in the embryos from each group.

## Natur-IVF blastocysts show fewer aberrantly expressed genes than C-IVF blastocysts

Natur-IVF and C-IVF blastocysts shared 334 genes that were aberrantly expressed in both groups vs In vivo (Exclusive DEGs, *Figure 3C*- *Figure 3—source data 2*). However, there were 440 genes (from the 784 DEGs in C-IVF) that showed aberrant expression only in C-IVF vs In vivo (DEGs only in C-IVF, *Figure 3C*), while 40% fewer genes (n = 281 from the 620 DEGs in Natur-IVF) showed aberrant expression only in the Natur-IVF group vs In vivo (DEGs only in Natur-IVF, *Figure 3C*). Importantly, several genes related to epigenetic reprogramming (down: *DNMT3B*, *DNMT1*; up: *HDAC5*, *KDM5A*), embryo development (down: *CTGF*, *ING2*, *KIT*, *EZH2*; up: *BMP4*, *TLN1*, *ADAR*), cell growth (down: *CDCA5*, *SMC1A*; up: *RB1*, *SMARCA2*) or imprinting (up: *IGF2BP2*, *GNAS*; down: *DIRAS3*) were amongst the C-IVF-specific DEGs (*Figure 3D*).

Direct comparison between Natur-IVF vs In vivo and C-IVF vs In vivo DEGs revealed that only 29 genes reached significant expression differences between the two in vitro groups after DESeq2 analysis (*Figure 3—source data 1*). Interestingly, of these 29 DEGs, 13 were similarly expressed in Natur-IVF and In vivo, and only seven showed similar expression between C-IVF and In vivo groups (*Figure 3C*, *Figure 3—source data 2*). Although the number of these genes was low, they could be critical because among the 13 genes exclusively different in the C-IVF blastocysts (*Figure 3—source data 2*), those down-regulated (n = 6) were *KIT*, *MPPA6*, *MTA3*, *KIF4A*, *UBR2* and *ISOC1* (Log Fold Change from −5.9 to −54.18). For all six genes data were available for the corresponding knock-out mice or knock-down studies, which showed phenotypes of altered/abnormal growth/size, reproduction/fertility, mortality/aging, hematopoietic system, homeostasis/metabolism and other abnormalities (*Supplementary file 2*).

These data suggest that in vitro culture significantly alters embryonic gene expression to a lesser extent than previously proposed (*Bauer et al., 2010*), and a better modulation of the blastocyst transcriptome was achieved by mimicking physiological conditions of fertilization and early embryo development by the addition of reproductive fluids (Natur-IVF).

## Genome-wide DNA methylation of the pig blastocyst is affected by the in vitro culture system

In this study, for the first time, whole-genome DNA methylation profiles on individual porcine blastocysts were generated by a low-cell adaptation of the post-bisulphite adaptor-tagging (PBAT) method (*Miura et al., 2012*; *Peat et al., 2014*). Three blastocysts from each group were analyzed. The number of unique alignments in the samples ranged from 13,150,508 to 42,208,651 and the coverage of CpGs ($\geq$1 read) from 52% to 59.2%. The global methylation percentages of CpGs were 15.02 ± 3.3, 11.09 ± 2.6 and 12.33 ± 3.6 for the C-IVF, Natur-IVF and In-vivo groups, respectively. The distribution of methylation levels in windows of 150 CpGs across the genome and a general view of the methylation profiles of the nine individual blastocysts are shown in *Figure 4A–B*. The generally low level of methylation suggests that the genome has experienced substantial loss of methylation from the gametes, analogous to that observed in other mammals (*Guo et al., 2014*; *Kobayashi et al., 2012*). The landscape of methylated cytosines suggests some structure across the genome, with regions with more methylation consistent between the individual blastocysts (*Figure 4B*). What contributes to this structure, for example, the regions of relatively higher

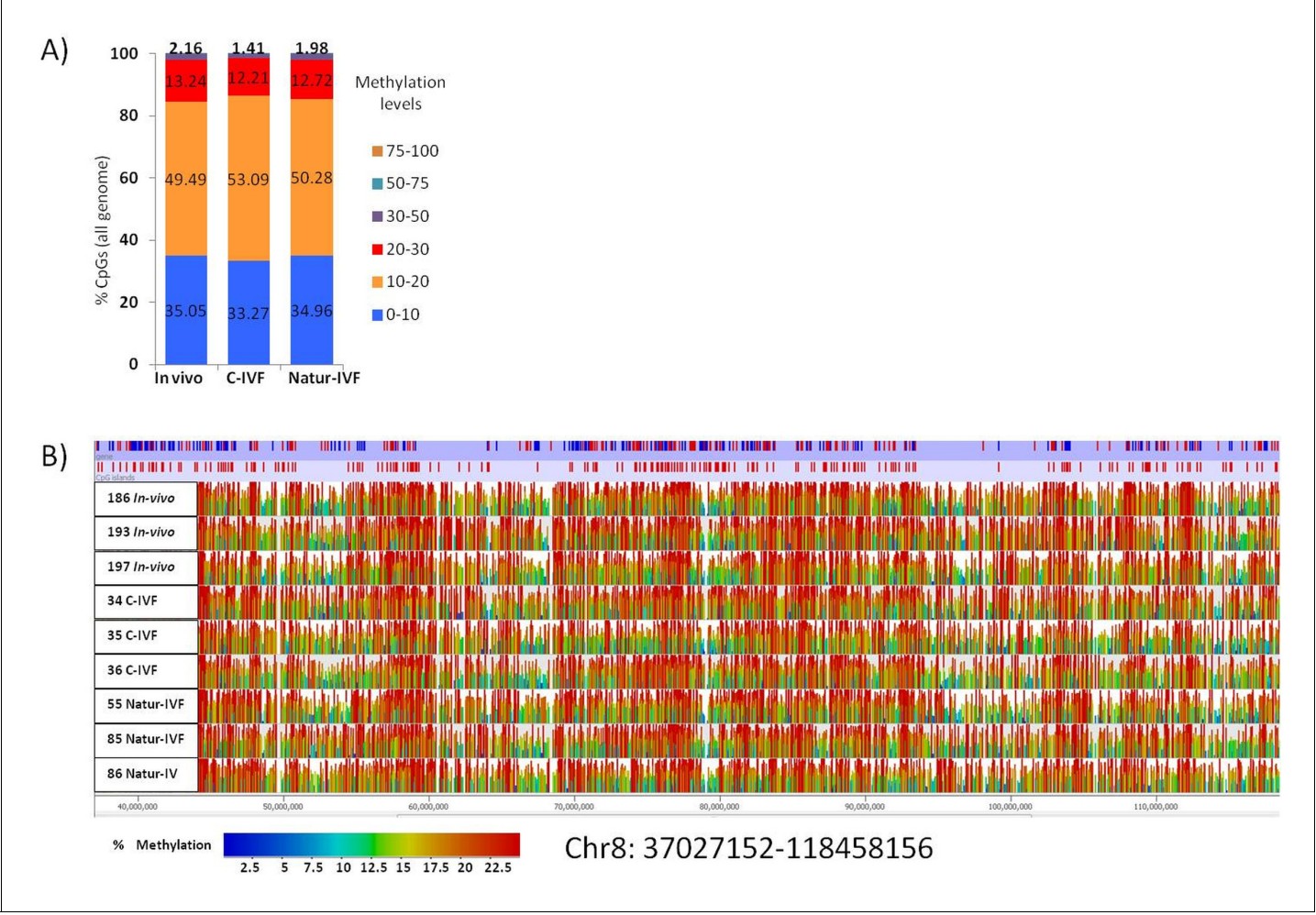

**Figure 4.** Distribution of methylation levels and general view of the methylation profiles of 9 individual pig blastocysts. (**A**) Distribution of methylation percentages across tiles of 150 CpGs on the pig genome for three groups of blastocysts (In-vivo, C-IVF and Natur-IVF). (**B**) Random browser shot as example of methylation landscape of the nine individual blastocysts analysed (Chr8:37027152–118458156). The two first rows in the picture represent the genes and CpG islands annotated (Ensembl, RRID:SCR_006773 *Sus scrofa* 10.2) in the pig genome, respectively. Color scale represents methylation levels from red (highest methylation, up to 25%) to blue (lowest methylation-0%).

methylation, is not immediately obvious, as methylation was similar in different genomic contexts with no marked enrichment in repetitive elements, for example (*Table 3*). Regarding the different classes of blastocysts, methylation over specific genomic features followed the same tendency as the global differences, with higher values for C-IVF (*Table 3*).

PCA revealed a good level of clustering for In-vivo and Natur-IVF embryos but not for C-IVF embryos (*Figure 5A*). In particular, embryos C34 and C36 were far from the other seven embryos analyzed.

The low level of global methylation suggested that few differentially methylated regions (DMRs) could be found. For this reason, and to obtain an unbiased measure of differences in genome methylation, a fixed size of 150 CpGs was used for analysis, as this was found to give a modal tile size of around 3 kb with about 150 reads per tile for most individuals. To make the data comparable to enable the detection of DMRs, separately from the global changes, the tiles informative in all samples (258,885) were extracted and quantile normalized. To identify DMRs, the comparison was filtered to require a consistent $\geq$5% absolute methylation change between all replicates of the first and second condition, followed by a T-test (B-H adjusted p<0.05). Differences between the groups were observed with fewer than 4000 DMRs for each pair-wise comparison (*Figure 5—source data*

**Table 3.** Percentages of methylation over genome features in porcine blastocysts produced in vitro (C-IVF and Natur-IVF) or collected in vivo (In vivo).

**% Methylation**

|  | In vivo | C-IVF | Natur-IVF |
|---|---|---|---|
| CpG islands | 9.69 | 11.80 | 10.11 |
| Promoters | 9.26 | 11.61 | 9.11 |
| TU | 12.84 | 15.47 | 12.36 |
| Intergenic | 11.75 | 14.48 | 11.37 |
| LINE1 | 12.63 | 15.43 | 12.02 |
| LTR | 12.77 | 15.53 | 12.06 |
| SINE | 12.45 | 15.30 | 11.94 |
| GLOBAL | 12.33 | 15.02 | 11.09 |

*1*). Globally, fewer DMRs showed higher methylation in In vivo vs Natur-IVF (n = 1,660) than in In vivo vs C-IVF (n = 2244) (*Figure 5B*).

To better characterize the changes in methylation exclusively affecting one of the groups (p<0.05 for both comparisons), the corresponding subsets of DMRs ('exclusive' DMRs for each group) were obtained by combining the previous lists (*Figure 5B,C,D and E*; *Figure 5—source data 2*), and the enrichment in specific features in those DMRs was evaluated (*Supplementary file 3*). For the three subsets of DMRs, there was a lower proportion of promoters compared to the global average (p<0.001). A lower proportion of LINE1s (p<0.05) was also found for the C-IVF group, while the Natur-IVF blastocyst group showed a higher proportion of DMRs in transcription units (defined over the annotated genes from 500 bp downstream of the annotated TSS, p<0.05). Both C-IVF and Natur-IVF DMRs were less enriched in intergenic regions (p<0.001) and at LTRs (p<0.05) than In vivo blastocysts. These departures from the methylation state might reflect global differences in the DNA methylation and/or demethylation capacity of the different groups at a developmental time when DNA methylation is rather dynamic.

Exclusive DMRs for each group were linked to Canonical Pathways (p<0.01) and Diseases and Bio Functions (adjusted p-value < 0.05; *Figure 6*) by IPA software. Representative genes for specific DMRs in each group are listed in *Supplementary file 4*. A DMR overlapping *IGF2R,* a gene directly related with the LOS in ruminants and mouse, was found in the subset of exclusive C-IVF DMRs (*Figure 5—source data 2*). The methylation percentages for this region (Chr1: 9,199,522–9,201,143) were 12.45%, 28.3% and 35.5% for C-IVF, Natur-IVF and In vivo, respectively (*Figure 7A*). In addition, a CpG island (oe = 0.89, Chr1:9,200,658–9,202,276) that overlapped the DMR showed significant differences in methylation (p<0.05): 14.1%, 27.8% and 29.4% for C-IVF, Natur-IVF and In vivo groups, respectively (*Figure 7B*), although we should be cautious about their significance since the CpG island distribution in the pig genome is very different to the human or mouse genome.

Top Diseases and Bio Functions linked by IPA to DMRs exclusive for each group with low or high methylation are represented in *Figure 6*. Top Molecular and Cellular Functions and representative genes related to DMRs with higher or lower methylation in each group (C-IVF, Natur-IVF and In vivo) are listed in *Supplementary file 4*.

## Three imprinted genes were differentially methylated in C-IVF, but not in Natur-IVF blastocysts, compared to in vivo blastocysts

Following the finding of a DMR at *IGF2R*, targeted analysis of candidate imprinted genes was done, as the differentially methylated regions of imprinted gene (igDMRs) are expected to maintain constant methylation in preimplantation embryos to ensure faithful imprinted expression of the associated genes throughout development. Therefore, they represent sites of methylation in preimplantation of clear biological significance. To identify putative igDMRs in the pig genome, all mouse igDMRs were lifted-over onto the pig genome. Where this was not possible, a gene-by-gene approach was taken to find the best possible fit for a candidate igDMR based on the known

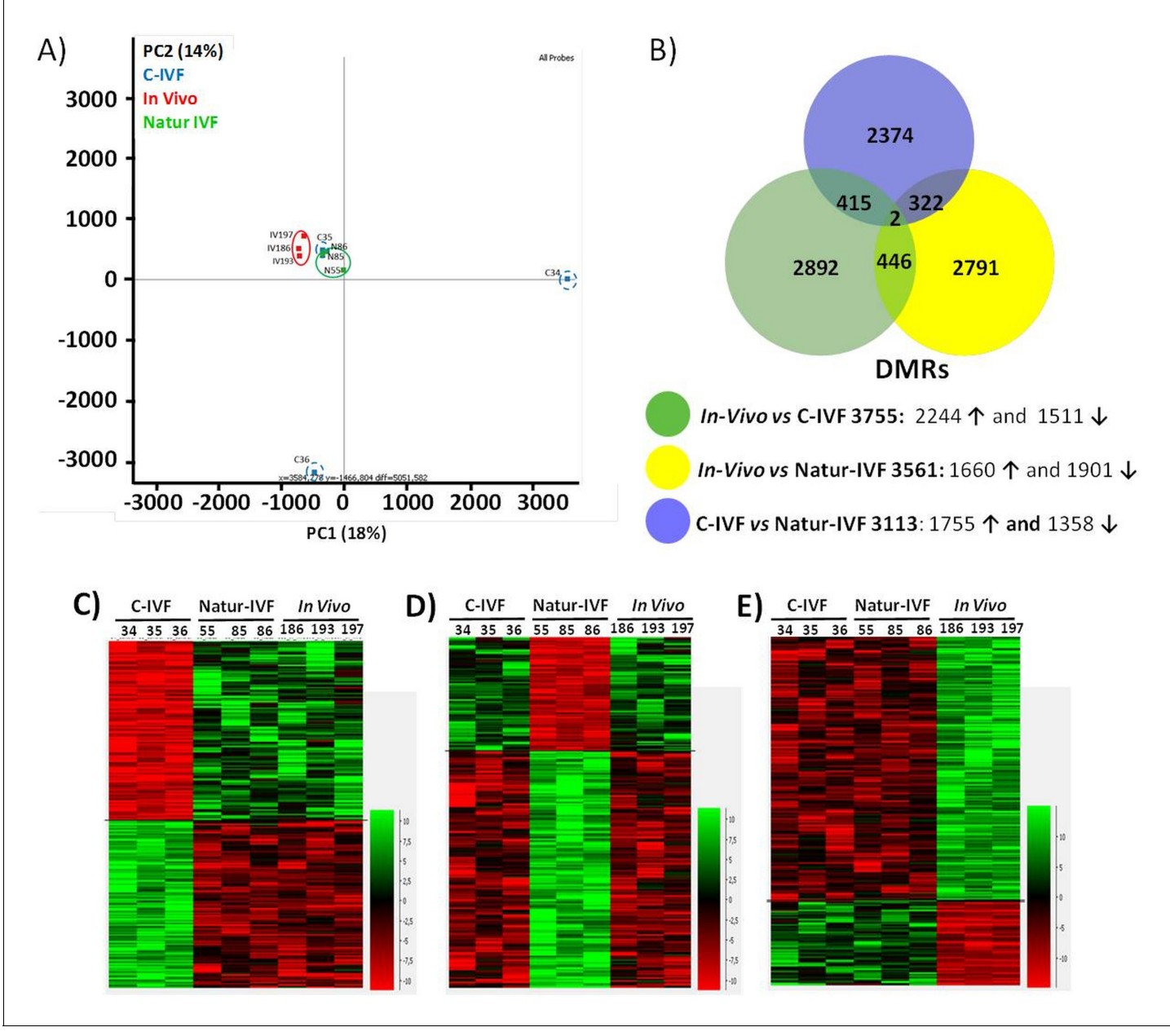

**Figure 5.** DNA-methylation analysis in blastocysts obtained in vivo, by the Natur-IVF system or by C-IVF system. (A) Principal Component Analysis (PCA) of the DNA methylation samples: In vivo embryos (red), Natur-IVF (green) and C-IVF (blue). Numbers denote ID of specific embryo. (B) Venn diagram of DMRs by pair-wise comparison (adjusted-p <0.05). Number of DMRs with higher (↑) or lower (↓) methylation in each pair-wise comparison are indicated (*Figure 5—source data 1*). (C) Heatmap of the 417 DMRs between the C-IVF group and the other two groups (In vivo and Natur-IVF). (D) Heatmap of the 324 DMRs between Natur-IVF group and the other two groups (In vivo and C-IVF). (E) Heatmap of the 448 DMRs between the In vivo group and the other two groups (Natur-IVF and C-IVF). For C, D and E (*Figure 5—source data 2*): Relative methylation measure as the difference in percent of methylation from the median methylation across all samples.

The following source data is available for figure 5:

**Source data 1.** All differentially methylated regions (DMRs) for each pair-wise comparison (C-IVF vs Natur-IVF, In vivo vs Natur-IVF, C-IVF vs In vivo).
**Source data 2.** Differentially methylated regions (DMRs) exclusive for each group (C-IVF, Natur-IVF, In vivo). This data related to *Figure 5B,C,D and E*.

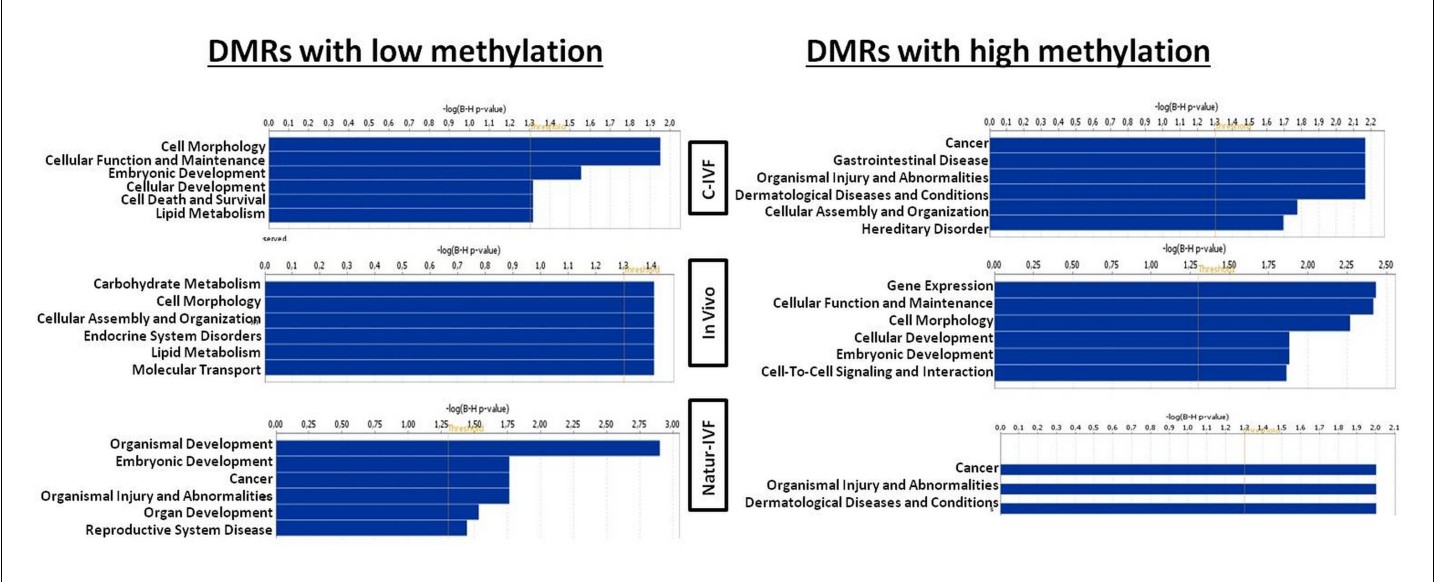

**Figure 6.** Top Diseases and Bio Functions linked by Ingenuity Pathways Analysis to DMRs exclusive for each group with low or high methylation.

organization of the corresponding mouse imprinted gene. All the genomic regions were then inspected manually to confirm that the correct regions had been found (*Table 4A*). It is not possible to conclude that all regions were actually igDMRs (as this would require methylation information from oocyte and sperm) and, indeed, the methylation values indicated that for some of the genes there was no conserved DMR (i.e. methylation in blastocysts was far below the theoretical 50%) and the associated locus was unlikely to be imprinted. This would seem to be the case, for example, for the genes *IMPACT*, *ZFP787* and *ZFP777*. For some, there was difficulty in finding possible homologous igDMRs, probably because of gaps in the porcine genome assembly (such as *SNRPN*, *KCNQ1* and *GRB10*), and there were a number of others that were excluded because the homologous pig region had no suggestion of a CpG island in the region equivalent to the igDMR in mouse (e.g. *U2AF1-RS1, MCTS2/H13*). Comparison of methylation in the three groups of blastocysts for the resulting 14 candidate igDMRs (with sufficient read coverage) revealed differences for *ZAC1* and *PEG10*, which were more methylated ($p < 0.05$) in the C-IVF than in In vivo group, and *PEG10* and *NNAT*, which were more methylated ($p < 0.05$) in the C-IVF than in Natur-IVF and In vivo groups (*Table 4B*). No statistical differences were found between Natur-IVF and In vivo groups. Of these three igDMRs, the one at *NNAT* coincides with the promoter CpG island (*Kobayashi et al., 2012*) and, in addition, one 150 CpG tile overlapping NNAT had methylation higher than 50% in C-IVF in the unbiased analysis (*Figure 8*).

## Discussion

The milieu in which fertilization and embryo development takes place is crucial for healthy fetal and offspring growth, as revealed by developmental and epigenetic alterations as a consequence of in vitro culture and ART (*Fernández-Gonzalez et al., 2004*; *Kleijkers et al., 2014*; *Lazaraviciute et al., 2014*; *Song et al., 2015*). However, the progress made by ART during the past two decades make a future without their use inconceivable, thus it is necessary i) to characterize the real epigenetic cost of ART, separated from other factors and ii) to develop new protocols to safeguard against possible negative impacts in offspring. Our study evaluated, by single blastocyst profiling, the genetic and epigenetic impacts of modified protocols to produce embryos in vitro that mimic, as far as possible, the physiological conditions of fertilization and early embryo development. This imitation of the natural environment was first approached in both gametes separately: in the male gamete, by using sperm selection procedures that avoided centrifugations, and sperm washing and processing media containing oviductal fluid from the pre-ovulatory phase of the cycle; and, in the female gamete, by

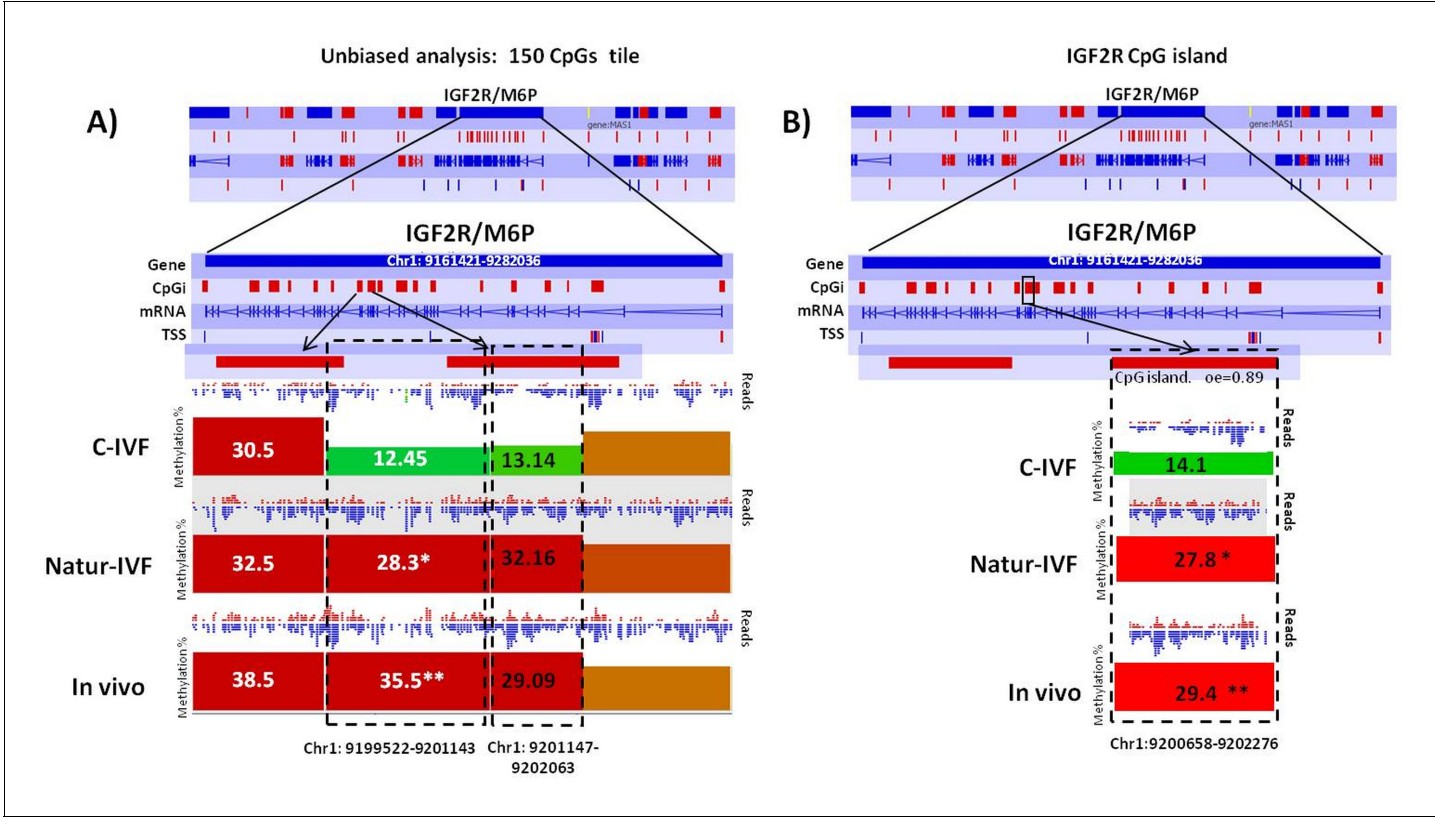

**Figure 7.** Methylation differences at IGF2R. (A) Methylation quantitation at *IGF2R* from the *unbiased analysis* of genome methylation in SeqMonk with a fixed size of 150 CpG windows. Mean percentages of methylation are shown by the bars for each group. Blue (unmethylated) and red (methylated) dots represent methylation reads. Asterisks indicate that methylation at the indicated region showed significantly different values (p<0.05) in Natur-IVF (*) and In vivo (**) vs C-IVF. TSS: transcription starting site. (B) Detailed view and methylation quantitation of the CpGi at the identified IGF2R DMR. Red rectangles represent, as indicated, CpG islands of the genes. Black boxes indicate the position of the targeted features, whose mean percentages of methylation are shown by the bars for each group. Blue (unmethylated) and red (methylated) dots represent methylation reads.

preincubating the oocytes within the precise fluid they encounter when, after ovulation, they are transported through the ampulla of the oviduct to the fertilization site, at the ampullar-isthmic junction (*Halbert et al., 1988*). Secondly, two experimental groups were established for a comparison with the in vivo specimens, where either BSA or reproductive fluids (obtained sequentially at the corresponding phases of the cycle) were added at every step of the IVF and EC procedures.

The results showed that reproductive fluids improve the outcome of IVF and the quality of pig blastocysts produced in vitro. The approach used, with spermatozoa coming from boars selected by their excellent reproductive performance, avoids the possibility of aberrations due to a paternal factor, which cannot be avoided in the human model, and helps to elucidate the epigenetic cost of ART independently of any paternal pathology. The figure of >40% progression of the cleaved embryos to blastocysts in vitro means an improvement over the best previous results (*Redel et al., 2016*). Nonetheless, the most remarkable findings were that Natur-IVF blastocysts attained a more advanced developmental stage and that the mean number of cells per blastocyst was the same as In-vivo embryos and 61% higher than C-IVF ones, which it is also above some of the best data previously reported in pigs (*Redel et al., 2016*). These results indicate that the use of reproductive fluids as additives, even at the low dose used in this study (1%) is beneficial for in vitro development of pig embryos so that it is now possible to obtain similar or even higher yields in the pig (45%) than in the bovine species. Although the possibility of transferring these methods to the human clinic might seem far off, the fact that nowadays other natural fluids such as breast milk for baby feeding or blood serum for transfusions are collected and stored at biobanks, make it possible to predict the future availability of human reproductive fluids obtained from oocyte donors during interventions at

**Table 4.** Targeted analysis of candidate imprinted genes. (A) Predicted imprinted regions in the pig genome by lifted-over mouse igDMRs the pig genome and manually inspected. (B) Pair-wise comparison of methylation by Analysis Chi-Square in the three groups of blastocysts for the resulting 14 candidate igDMRs. *C-IVF vs In vivo: p<0.05 with 20 minimum observations and 10 minimum percentage of difference % methylation. ** C-IVF vsNatur-IVF: Analysis Chi- Square p<0.05 with 20 minimum observations and 10 minimum percentage of difference % methylation. Natur-IVF vs In vivo: no statistical differences.

A)

| Tile | Chromosome | Start | End |
|------|-----------|-------|-----|
| IGF2R/AIR | 1 | 9,244,239 | 9,248,054 |
| ZAC1 | 1 | 23,638,887 | 23,643,228 |
| SOCS5 | 3 | 99,885,360 | 99,887,132 |
| ZFP787 | 6 | 55,574,080 | 55,575,926 |
| ZIM2 | 6 | 56,641,190 | 56,644,823 |
| IMPACT | 6 | 102,001,929 | 102,002,533 |
| NAT1l5 | 8 | 139,773,830 | 139,775,461 |
| PEG10 | 9 | 81,642,957 | 81,644,146 |
| INPP5FV2 | 14 | 141,186,219 | 141,188,231 |
| NNAT | 17 | 46,041,843 | 46,045,629 |
| NESPAS | 17 | 66,313,673 | 66,320,932 |
| GNAS-exon1a | 17 | 66,348,009 | 66,352,062 |
| MEST | 18 | 19,340,335 | 19,345,549 |
| ZFP777 | 18 | 60,941,421 | 60,943,096 |

B)

| Tile | Chromosome | C-IVF | Natur-IVF | In-vivo |
|------|-----------|-------|-----------|---------|
| ZAC1 | 1 | 42.41 | 33.55 | 23.87* |
| PEG10 | 9 | 47.75 | 36.91** | 30.90* |
| NNAT | 17 | 34.63 | 19.22** | 23.28* |

human infertility clinics (*Coy and Yanagimachi, 2015*). In fact, the first samples of these fluids are already stored at Biobanc-Mur in Spain (National Register of Biobanks N° B.0000859).

Our study also showed that Natur-ART blastocysts are closer to the gene expression profile of the In vivo blastocysts than C-IVF blastocysts. Amongst the most striking differences found was the expression of genes related to epigenetic reprogramming. It has been shown in mice and human that during the transition from zygote to blastocyst there is a massive loss of DNA methylation, with the exception of imprinted genes and some repetitive elements (*Guo et al., 2014*; *Reik and Kelsey, 2014*). In agreement with this observation, the global methylation level in the three groups of pig blastocysts analyzed was below 15%, suggesting that they had largely undergone a reprogramming event. This globally low methylation level compared to somatic cells or gametes, made it difficult to find high quantitative differences between embryos. Despite this, methylation percentage was higher in C-IVF embryos than in the other two groups, in agreement with previous studies indicating that ART-derived blastocysts displayed higher levels of methylation than in vivo derived ones (*Deshmukh et al., 2011*). This difference appeared to be global, with all features affected and, no evidence of multiple sub-groups over different genomic regions; therefore, there was no indication of specific regions resisting reprogramming. At the same time, genes for DNMT1 and the binding protein of its crucial cofactor UHRF1, which are considered responsible for maintenance of methylation patterns in replicating DNA and for maintaining imprints during preimplantation embryonic stages, were less expressed in C-IVF blastocysts, as was *DNMT3B*, required for de novo

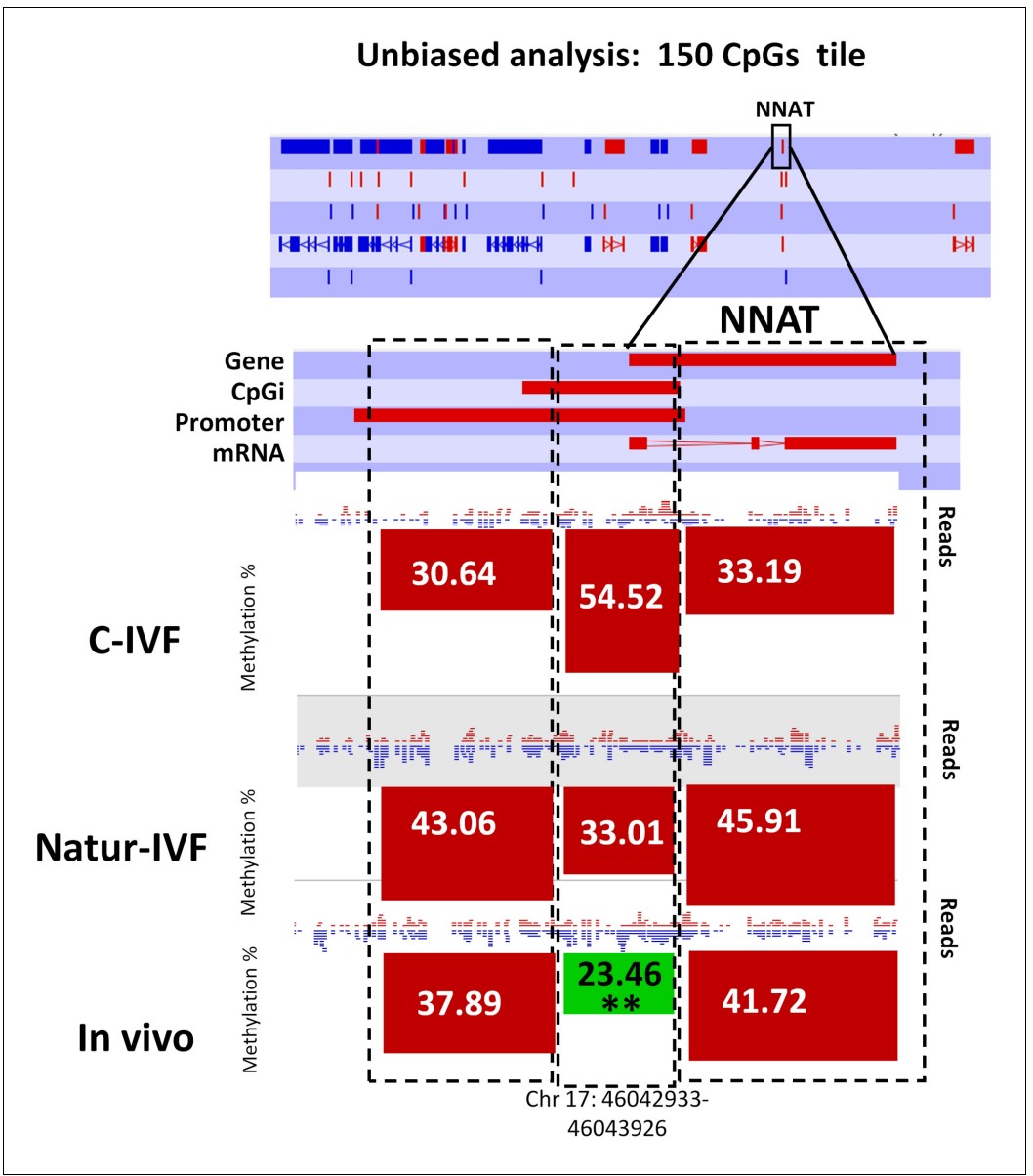

**Figure 8.** Methylation quantitation at *NNAT* from the *unbiased analysis* of genome methylation in SeqMonk with a fixed size of 150 CpG windows. Black boxes indicate the position of the selected 150 CpG windows, whose mean percentages of methylation are shown by the bars for each group. Blue (unmethylated) and red (methylated) dots represent methylation reads. Asterisks indicate that methylation at the indicated region (black box) showed significantly different values (p<0.05) in Natur-IVF (*) and In-vivo (**) vs C-IVF.

remethylation from this stage onwards. Differences in cell numbers, as a result of a probable additional round of cell division in In vivo and Natur-IVF embryos compared to C-IVF, is unlikely to explain a shift from ~11–12% to ~15% global methylation. All together, these data suggest an impaired demethylation in the C-IVF group. Analysis of hemimethylated CpG dyads by deep hairpin bisulfite sequencing, as recently reported in mouse (*Arand et al., 2015*), could help to clarify this issue.

A second key finding in this study was that the methylation levels in the samples analyzed showed much lower overall methylation levels (mean across all samples was 13.1%) than would be expected from somatic tissues. Furthermore, there were differences in the global mean methylation levels between different samples, ranging from 8.9% to 18.5%. Taken together, these observations

suggest that the samples were collected during a time of global methylation reprogramming. The variability in global methylation levels would have confounded a direct comparison focussing on locus-specific methylation differences, so to account for this a quantile normalization was required to allow for a direct quantitative comparison of methylation levels.

Given that these samples are undergoing active reprogramming, it is also not unreasonable to think that some previously reported DMRs may not be established yet, or that the strength of the DMRs would be reduced. Despite this, we were able to find candidate DMRs between the groups with a reasonable statistical significance, although the magnitude of the methylation differences was low. Considering that previous studies have shown extremely close correlations between qPCR and RNA-seq data (*Asmann et al., 2009*; *Griffith et al., 2010*; *Wu et al., 2014*) and that validation by qPCR has its own probe-bias based on what region of the cDNA is amplified, we deem, in contrast to microarrays data, that there is not solid evidence that validation of the RNA-Seq and DNA methylation results by qPCR will provide extra significance to our results. For this reason, we did not perform qPCR validation in this study.

Another key observation in this study was that the in vitro culture affects imprinted gene expression and methylation. Plasticity of the preimplantation embryo could enable a recovery of alterations in methylation and further expression of non-imprinted genes during development, but any erosion of methylation marks at imprinted genes are unlikely to be corrected. In our data, from the 10 candidate imprinted regions retaining more than 30% of methylation in the pig blastocysts, we found three in C-IVF (*ZAC1*, *PEG10* and *NNAT*) with significantly different methylation compared to In vivo blastocysts, and two (*PEG10* and *NNAT*) compared to Natur-IVF. Knock-out mice lacking *PEG10* showed early embryonic lethality with placental defects, indicating the importance of this gene in embryonic development (*Ono et al., 2006*). The protein encoded by *NNAT*, on the other hand, may be involved in the regulation of ion channels during brain development and may also play a role in forming and maintaining the structure of the nervous system. Defects in methylation at *ZAC1* and *IGF2R* have been found in patients with the imprinted disorders transient neonatal diabetes mellitus (TNDM) or Silver-Russell syndrome (SRS), respectively, including those born following the use of ART (*Le Bouc et al., 2010*). In addition, genes related to the IGF axis, *IGF2BP2* and *IGF2BP2-IMP2*, were up-regulated in C-IVF, and *IGF2R* in both C-IVF and Natur-IVF embryos. Altered *IGF2BP2* expression in C-IVF is of interest, since reduced abundance of IGF2 has been associated with lower fetal weight after in vitro culture (*El Hajj and Haaf, 2013*). The imprinting status of *IGF2R* in the pig is unclear (*Killian et al., 2001*; *Braunschweig, 2012*) but, independently of this uncertainty, our data indicated higher expression of this gene in the two in vitro groups of blastocysts, which would be in agreement with previous reports in other species and could indicate a possibility of LOS-related alterations observed in abnormal in vitro and cloned embryos (*Young et al., 2001*). At the same time, the reduced methylation in *IGF2R* specifically in the C-IVF group could suggest that this group is more likely to be susceptible to sustained deregulation of *IGF2R* expression and a greater probability of LOS-like syndromes.

Altered expression in both groups of blastocysts produced under in vitro conditions was observed in some genes related to embryonic development, but some aberrations were absent in Natur-IVF embryos. In human blastocysts, it has been observed that those with higher implantation rate and higher number of cells per embryo showed up-regulation of *DNMT3A* (*Kleijkers et al., 2015*). In our data, the In vivo and Natur-IVF blastocysts showed a higher number of cells than those from the C-IVF group, in which expression of *DNMT3A* was decreased. We also observed higher expression of *CDKN1A* in the two in vitro groups, with an intermediate value in Natur-IVF. CDKN1A inhibits embryonic cell proliferation in response to DNA damage and it is considered one of the key genes responsible for the abnormalities in ART embryos since an aberrant increase of CDKN1A expression might be related to the growth-defect phenotype (*Ishimura et al., 2016*). Methylation of the *CDKN1A* gene, however, was similar in all three groups, between 5 and 7%. Other genes involved in DNA repair and cell cycle regulation were found to be altered, such as *MDM2* (in C-IVF) and *TP53INP* (up-regulated in Natur-IVF and C-IVF) and *HSPA4L, HSP40B1, HSPH1, HSP90* (down-regulated only in C-IVF). Altered expression of these genes may limit the ability of the embryo to respond to DNA damage, such that in vitro culture may lead to dysregulation of such genes, thus affecting long-term embryo viability (*Zheng et al., 2005*). The same situation was found for *SLC2A3* (Glut-3) and *SLC2A2*, which have been related to LOS (*Wrenzycki et al., 2004*) and were highly up-regulated in the two in vitro groups. Again, no differences at the methylation level were found for

any of these genes. Although DNA methylation at the promoter/gene bodies is directly/indirectly correlated with gene expression, this is not strictly true during the periods of dramatic loss of DNA methylation, as occurs during early embryo development or primordial germ cells (PGC) formation. For example, *Gkountela et al. (2015)* showed a general uncoupling between DNA methylation and gene expression during demethylation of PGCs, commenting 'Our data reveal a remarkable and pervasive loss of DNA methylation in human PGCs and AGCs during prenatal life that has almost no relationship to changes in gene expression'. Comparative analyses between our methylation and gene expression data also showed this lack of correlation. In our opinion, at this stage of development and with this low level of methylation, this was an expected result.

Finally, the exclusive alteration in C-IVF of genes such as *KIT*, whose knock-out in mouse results in multiple alterations including embryonic lethality (*Ro et al., 2010*), *UBR2*, whose deletion results in female embryonic lethality and growth arrest (*Kwon et al., 2003*), or *ISOC1*, whose mutation produces phenotypes with body weight loss (*Rainger et al., 2013*), support the hypothesis that offspring produced with Natur-IVF conditions would be healthier than those produced with C-IVF, although additional studies are necessary to confirm this finding.

In conclusion, we report here the first time genome-wide DNA methylation and transcription analysis in single blastocysts (in vivo and in vitro) of a mammalian species and propose a new strategy for prevention of aberrant epigenetic and gene expression profiles induced by ART. This strategy, based on the addition of reproductive fluids in the culture media used during the ART procedures, can be applied in other animals as well as in humans, after safety concerns of transmission of diseases have been properly addressed. The design of new culture media containing all the proteins that are naturally present in the original biological fluid, represents not only a technical challenge but a biomedical responsibility that must be addressed to prevent future pathologies both in animals and humans. In addition, we offer a new protocol for the in vitro production of pig embryos with a significant improvement over the previous data published. Our study represents a new form of thinking in the field, far from the chemically defined culture media, and could help to face one of the biggest milestones of the current reproductive medicine: safer ART.

## Materials and methods

### Culture media
Unless otherwise indicated, all chemicals and reagents were purchased from Sigma-Aldrich Quimica S.A. (Madrid, Spain).

### Collection and processing of follicular, oviductal and uterine fluids
Fluids were obtained from animals raised at a commercial farm (CEFU, S.A., Murcia, Spain) and slaughtered in an abattoir belonging to a food industry (El Pozo, S.A) near the University of Murcia. For the collection of follicular fluid, ovaries from 6-month-old Large White animals weighing 100–110 kg were transported to the laboratory in saline containing 100 µg/ml kanamycin sulfate, washed once in 0.04% cetrimide solution (alkyltrimethylammoniumbromide) and twice in saline within 30 min of slaughter. The content of follicles between 3 and 6 mm diameter, from at least 50 ovaries (25 females), was quickly aspirated, centrifuged at 1800 *g* for 30 min at 4°C and the supernatant filtered through 0.22 µm diameter filter (*Naito et al., 1988*). One ml follicular fluid (FF) aliquots were stored at −80°C until their use as additives for the IVM medium.

For the collection of oviductal (OF) and uterine (UF) fluids, genital tracts from cyclic Large White sows (2–4 years old) were obtained at the abattoir and transported to the laboratory on ice within 30 min of slaughter. The cyclic stage of animals was assessed once in the laboratory, on the basis of ovarian morphology on both ovaries from the same female. Oviducts and uteri were classified as early follicular, late follicular, early luteal or late luteal phase (*Carrasco et al., 2008*). Both oviducts and uteri coming from the same genital tract were classified as in the same stage of the cycle. Once classified, oviducts and uteri were separated and quickly washed once with 0.4% v/v cetrimide solution and twice in saline. Oviducts and uteri were dissected on Petri dishes or trays, respectively, sitting on ice. Once dissected, OF were collected by aspiration with an automatic pipette by introducing a 200 µl pipette tip into the ampulla and manually making an increasing pressure gradient from the isthmus to the ampulla. The UF was collected by making a manual increasing pressure

gradient from the proximal end to the distal end (utero-tubal junction) of the uterine horn and letting the fluid drop into a sterile 50ml Falcon tube. Once recovered, samples (OF and UF) were centrifuged twice at 7000 $g$ for 10 min at 4°C to remove cellular debris. Then the supernatant was immediately stored at −80°C until use. Oviducts from animals at the late follicular phase (POF-LF) and at the early luteal phase (POF-EL) gave a mean volume of around 50 μl and 40 μl, respectively per oviduct. At the early luteal phase, approximately 10 ml of UF per uterine horn were collected each time. Aliquots of 50 μl OF and 50 ml UF of pooled samples from at least 20 animals for OF and five animals for UF were used. Only samples that passed quality controls (pH 7.0–7.6, osmolality 280–320 mOsm/kg, endotoxin <0.10 EU/mL, a minimum 90% of Metaphase II oocytes after IVM with FF and ZP hardening for oocyte preincubation in POF-LF >1 hr) were used for experiments.

## Oocyte collection and in vitro maturation

Ovaries from 6 months old animals weighing 100–110 kg were transported to the laboratory in saline containing 100 μg/ml kanamycin sulfate at 38°C, washed once in 0.04% cetrimide solution and twice in saline within 30 min of slaughter. Cumulus–oocyte complexes (COCs) were collected from antral follicles (3–6 mm diameter), washed twice with Dulbecco's PBS (DPBS) supplemented with 1 mg/ml polyvinyl alcohol (PVA) and 0.005 mg/ml red phenol, and twice more in maturation medium previously equilibrated for a minimum of 3 hr at 38.5°C under 5% $CO_2$ in air. Maturation medium was NCSU37 supplemented with 0.57 mM cysteine, 1 mM dibutyryl cAMP, 5 mg/ml insulin, 50 μM $\beta$-mercaptoethanol, 10 IU/ml equine chorionic gonadotropin (eCG; Foligon; Intervet International BV, Boxmeer, Holland), 10 IU/ml human chorionic gonadotropin (hCG; Veterin Corion; Divasa Farmavic, Barcelona, Spain), and 10% porcine follicular fluid (v/v). Only COCs with complete and dense *cumulus oophorus* were used for the experiments. Groups of 50 COCs were cultured in 500 μl maturation medium for 22 hr at 38.5°C under 5% $CO_2$ in air. After culture, oocytes were washed twice in fresh maturation medium without dibutyryl cAMP, eCG and hCG and cultured for an additional period of 20–22 hr.

## In vitro fertilization

Before IVF, mature oocytes were preincubated in 100% porcine oviductal fluid (POF) from the late follicular (LF) phase (NaturARTs POF-LF) for 30 min (*Coy et al., 2008*) and then washed three times in TALP medium. TALP medium consisted of 114.06 mM NaCl, 3.2 mM KCl, 8 mM Ca-lactate.5H$_2$O, 0.5 mM MgCl$_2$.6H$_2$O, 0.35 mM NaH$_2$PO$_4$, 25.07 mM NaHCO$_3$, 1.85 mM Na-lactate, 0.11 mM Na-pyruvate, 5 mM glucose, 2 mM caffeine, 1 mg/ml PVA and 0.17 mM kanamycin sulfate. Either 3 mg/ml BSA-FAF (A-6003) or 1% of NaturARTs POF-LF was included as additives in the IVF medium for the first 8 hr of coincubation (C-IVF and Natur-IVF groups, respectively). Ejaculated spermatozoa from boars of proven fertility (1–2 years old) were transported to the laboratory and 1 ml of semen was lay below 1 ml of NaturARTs PIG sperm swim up medium (http://embryocloud.com) at the bottom of a conical tube. After 20 min of incubation at 37°C (with the tube at a 45° angle), 0.75 ml from the top of the tube were aspirated and used for insemination of the IVF dishes ($10^5$ cells/ml) with the oocytes. For the density gradient group, aliquots of the semen samples (0.5 ml) were centrifuged (700 $g$, 30 min) through a discontinuous Percoll (Pharmacia, Uppsala, Sweden) gradient (45% and 90% v/v) and the resultant sperm pellets were diluted in TALP medium and centrifuged again for 10 min at 100 $g$. Finally, the pellet was diluted in TALP and 250 μl of this suspension were added to the wells containing the oocytes, giving a final concentration of $10^5$ cells/ml.

Spermatozoa and oocytes were incubated at 38.5°C under 5% $CO_2$ for 8 hours. Later on, the putative zygotes were transferred to embryo culture medium. At this point, a sample of the putative zygotes from each group was collected, fixed and stained as previously described (*Coy et al., 2008*) to assess the fertilization rates (percentage of penetrated oocytes, percentage of monospermy, mean number of spermatozoa penetrating each oocyte and mean number of spermatozoa attached to the zona pellucida). Penetration rate was defined as the proportion of oocytes penetrated by one or more spermatozoa.

## In vitro culture of putative zygotes

Media for embryo culture were NCSU23 supplemented with sodium lactate (5 mM), pyruvate (0.5 mM) and non-essential amino acids (NCSU23a, for the first 48 hr) or NCSU23 supplemented with

glucose (5.5 mM) and essential and non-essential amino acids (NCSU23b, 48–180 hr). At 8 hr post insemination (hpi), putative zygotes were transferred to culture dishes containing NCSU23a medium and 0.4% BSA in the C-IVF group or 1% POF from the early luteal (EL) phase of the estrous cycle (NaturARTs POF-EL) in the Natur-IVF group. At 48 hpi, the cleavage was assessed under the stereomicroscope and the 2–4 cell stage embryos were transferred to NCSU23b with 0.4% BSA (C-IVF group) or 1% of porcine uterine fluid (PUF) from early luteal phase (NaturARTs PUF-EL, Natur-IVF group). On day 7.5 (180 hpi), blastocyst stage morphology was assessed under the stereomicroscope and later on a sample was fixed and stained (*Coy et al., 2008*) and the remaining blastocyst were washed in PBS and frozen in PCR tubes in the minimum volume of medium. The parameters assessed in the stained blastocysts were development stage (2–4 cells, 8–16 cells, morula or blastocyst), mean number of cells per blastocyst, and ability for hatching (rhythmic movements of expansion and contraction before going out of the zona pellucida). The blastocysts frozen for genetic and epigenetic study were passed through liquid nitrogen vapours for 5 s and immediately introduced in the freezer at -80°C until the day of use for RNA extraction or bisulphite treatment.

## Statistical analysis of IVF data

Data are presented as mean ± SEM, and all percentages were modeled according to the binomial model of variables and arcsin transformation to achieve normal distribution. The variables in all the experiments were analyzed by one-way or two-way ANOVA. When ANOVAs revealed a significant effect, values were compared by the Tukey test. A pvalue < 0.05 was taken to denote statistical significance.

## Collection of blastocysts In vivo

Ten sows 18-month old were weaned 21 days after second parturition and five days later showed signs of standing estrous. Animals were inseminated in the collaborative farm and slaughterhoused 7.5 days after. Genital tracts were collected and transported to the laboratory where uterine horns were briefly dissected and washed with PBS within 2 hr from slaughtering. Blastocysts were identified under the stereomicroscope, collected and immediately frozen as described for the in vitro produced embryos. A portion of these blastocysts was fixed in glutaraldehyde and stained with Hoechst for cell counting.

## Experimental groups

C-IVF group (C-IVF): six blastocysts classified as 7A according to Bo and Mapletoft (*Bo and Mapletoft, 2013*) (#34, 35, 36, 93, 94 and 96) were produced in vitro with BSA as the only protein source. Sperm were processed by swim up in NaturARTs sperm medium with BSA (Swim-up-BSA). IVF medium consisted of TALP (0–8 hr) and embryo culture medium consisted of NCSU23a (8–48 hr) and NCSU23b (48–180 hr). Natur-IVF group: six blastocysts classified as 7A (#55, 85, 86, 27, 54 and 60) were produced in vitro with NaturARTs POF and PUF as the protein source. Sperm were processed by swim up in NaturARTs sperm medium with NaturARTs POF-LF (Swim-up-Fluid). IVF medium consisted of TALP +1% NaturARTs POF-LF (0–8 hr) and embryo culture medium consisted of NCSU23a + 1% NaturARTs POF-EL (8–48 hr) and NCSU23b + 1% NaturARTs PUF-LL (48–180 hr). For both groups, before IVF oocytes were pre-incubated for 30 min in preovulatory oviductal fluid (NaturARTs POF-LF). In vivo group: six blastocysts classified as 7A (#186, 193, 197, 189, 190 and 191) were collected by flushing the uteri of animals within 2 hr of slaughtering. The animals were under natural heat after weaning and insemination was performed 7 days before slaughtering.

## RNA isolation and RNA-Seq

ARCTURUS PicoPure RNA Isolation Kit (KIT0204, Life Technologies) was used to extract the RNA from individual blastocysts. RNA-Seq libraries were generated using Ovation RNA-Seq System V2 (NuGEN, Cat. 7102–08) for low amount of starting material and further amplified with NEB Next DNA Library Prep Master Mix for Illumina (NEB, Cat. E6040S). All steps were performed according to manufacture guidelines. iPCRTag reverse primer with individual index was used to generate three independent biological replicates from each condition. 100 bp single end reads were sequenced on Illumina HiSeq 1000. Sequencing data were processed. For RNA-Seq libraries, raw sequence reads were trimmed using Trim Galore to remove adapter contamination and reads with poor quality

defined by low PHRED score. Mapping was performed using Tophat software (http://tophat.cbcb. umd.edu/) and data were visualized with Seqmonk (RRID:SCR_001913, http://www.bioinformatics. babraham.ac.uk/projects/seqmonk/). RNA quality was assayed by Bioanalyzer and even though each sample came from a single blastocyst, RIN score was between 6.1–8.2.

### Analysis of RNA-Seq data

Annotated pig mRNA features were quantitated with raw read counts in SeqMonk and these were fed into DESeq2 for differential expression analysis using a p-value cutoff of 0.05 and not applying independent filtering. Reads were subsequently re-quantitated as log2RPM (reads per million reads of library) and globally normalized to the 75th percentile of the data. Significant effect sizes were selected using the Seqmonk intensity difference filter where the difference in expression in each gene was compared to the set of differences in the 1% of the data with the most similar average expression level as the gene being tested. Only genes with significantly higher changes (p<0.05 after Benjamini and Hochberg correction) were kept.

### Bisulfite sequencing based on post-bisulfite adapter tagging

An adaptation of whole genome bisulfite sequencing that involves post-bisulfite adapter tagging (PBAT) (*Miura et al., 2012*) was used to analyze the methylome of individual pig blastocysts at single-base resolution on a genome-wide scale. Further modification of the method described in Smallwood *et al.* (*Smallwood et al., 2014*) was used to generate BS-seq libraries. Briefly, an individual blastocyst was lysed for 1 hr in 1% SDS with proteinase K and treated with bisulfite reagent using Imprint DNA modification kit (Sigma, MOD50). DNA was eluted in EB buffer and one round of first strand synthesis was performed using a biotinylated oligo 1 (5-[Btn]CTACACGACGCTCTTCCGATC TNNNNNNNNNN-3). Samples were further treated with Exonuclease I, washed and eluted in 10 mM Tris-Cl and incubated with washed M-280 Streptavidin Dynabeads (Life Technologies) to pull down the biotinilated fraction of DNA. Second strand synthesis was performed using oligo 2 (5'-TGC TGAACCGCTCTTCCGATCTNNNNNNNNN −3') and samples were amplified for 15 PCR cycles using indexed iPCRTag reverse primer (*Smallwood et al., 2014*) with KAPA HiFi HotStart DNA Polymerase (KAPA Biosystems) and purified using 0.8× Agencourt Ampure XP beads (Beckman Coulter). Libraries were assessed for quality and quantity using High-Sensitivity DNA chips on the Agilent Bioanalyser, and the KAPA Library Quantification Kit for Illumina (KAPA Biosystems). Three libraries generated from individual blastocysts for each experimental condition were prepared for 100 bp single-end sequencing on Illumina HiSeq 1000 and sequenced at three samples per lane.

### Analysis of methylation data

For the unbiased analysis, tiles were defined in SeqMonk using the read position tile generator tool and selecting one read count per position and 150 valid positions per window, in all the nine individual data sets (286,136 tiles). Then, the bisulfite quantitation pipeline was run over existing tiles, one minimum count to include position and 20 minimum observations to include feature. To remove the tiles without data, the filter on values for individual tiles was applied, where values had to be between 0 and 100 for exactly 9 of the nine selected data stores. Then, tiles with data for all the samples were obtained (N = 258,885 tiles). Bisulphite quantitation pipeline was run again over the new tiles and data were normalized by the match distribution quantile normalization tool. Finally, every pair-wise comparison was filtered to require a consistent 5% change between all replicates of the first and second condition, and then replicate sets stats was applied where every comparison had a significance below 0.05 after Benjamini and Hochberg correction. For the targeted analysis of the candidate imprinted regions a Chi-Square test (p<0.05) was applied for every comparison.

## Acknowledgements

The authors thank CEFU, SA and El Pozo, SA for providing the biological material; Juan Antonio Carvajal and Soledad Rodriguez for collecting the oviducts, uteri and ovaries at the slaughterhouse; Carmen Matás for technical support with IVF and Kristina Tabbada for sequencing RNA-Seq and BS-seq libraries. **Funding:** Work in GK's laboratory was supported by the UK Biotechnology and Biological Sciences Research Council and Medical Research Council. Work in PC's laboratory was supported by grants AGL2012–40180 C03-01 and AGL2015–66341-R from the Ministry of Economy and

Competitiveness (Spain), and 20040/GERM/16 from Fundación Séneca. PC stay at The Babraham Institute was funded by a mobility grant of the Spanish Ministry of Education, Culture and Sports (PRX14/00348).

## Additional information

### Funding

| Funder | Grant reference number | Author |
| --- | --- | --- |
| Research Councils UK | | Gavin Kelsey |
| Ministerio de Economía y Competitividad | AGL2012-40180-C03-01 | Pilar Coy |
| Ministerio de Educación, Cultura y Deporte | PRX14/00348 | Pilar Coy |
| Fundación Séneca. Región de Murcia. Spain | 20040/GERM/16 | Pilar Coy |
| Ministerio de Economía y Competitividad | AGL2015-66341-R | Pilar Coy |

The funders had no role in study design, data collection and interpretation, or the decision to submit the work for publication.

### Author contributions

SC, Data curation, Formal analysis, Investigation, Methodology, Writing—original draft, Writing—review and editing; EI, Data curation, Validation, Methodology, Writing—review and editing; RR, Data curation, Validation, Investigation, Methodology, Writing—review and editing; SG-M, CS-Ú, HS, Data curation, Methodology; FAG-V, Data curation, Writing—review and editing; SA, Software, Formal analysis, Validation, Methodology, Writing—review and editing; GK, Supervision, Funding acquisition, Methodology, Writing—review and editing; PC, Conceptualization, Data curation, Supervision, Funding acquisition, Investigation, Methodology, Writing—original draft, Writing—review and editing

### Author ORCIDs

Sebastian Canovas, http://orcid.org/0000-0003-4190-6258
Gavin Kelsey, http://orcid.org/0000-0002-9762-5634
Pilar Coy, http://orcid.org/0000-0002-3943-1890

### Ethics

Animal experimentation: This study was carried out in strict accordance with the recommendations in the Guiding Principles for the Care and Use of Animals (DHEW Publication, NIH, 80-23). The protocol was approved by the Ethical Committee for Experimentation with Animals of the University of Murcia, Spain (Project Code: 192/2015).

## Additional files

### Supplementary files

• Supplementary file 1. Top Canonical Pathways, Physiological Systems and Molecular and Cellular Functions related to DEGs between blastocysts produced in vitro under two different systems.

• Supplementary file 2. Functions associated with the down-regulated genes in porcine blastocysts produced without reproductive fluids (C-IVF), compared to blastocyts produced using Natur-IVF system or collected in vivo.

• Supplementary file 3. Percentages of specific features included in the 150 CpG size DMRs exclusive for each of three groups.

• Supplementary file 4. Top Molecular and Cellular Functions and representative genes related to DMRs with higher or lower methylation in each group (C-IVF, Natur-IVF and In vivo).RNA-Seq and DNA methylation data available from the Dryad Digital Repository: 10.5061/dryad.n77r3

## Major datasets

The following dataset was generated:

| Author(s) | Year | Dataset title | Dataset URL | Database, license, and accessibility information |
|---|---|---|---|---|
| Canovas S, Ivanova E, Romar R, García-Martínez S, Soriano-Úbeda C, García-Vázquez FAA, Saa-deh H, Andrews S, Kelsey G, Coy P | 2016 | Data from: DNA methylation and gene expression changes derived from assisted reproductive technologies can be decreased by reproductive fluids | http://dx.doi.org/10.5061/dryad.n77r3 | Available at Dryad Digital Repository under a CC0 Public Domain Dedication |

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
