## [Decision Letter]

Editors’ note: a previous version of this study was rejected after peer review, but the authors submitted for reconsideration. The first decision letter after peer review is shown below.]

Thank you for submitting your work entitled "Epigenetic and gene expression changes derived from assisted reproduction can be mitigated by reproductive secretions" for consideration by *eLife*. Your article has been reviewed by three peer reviewers, and the evaluation has been overseen by a Reviewing Editor and a Senior Editor. The reviewers have opted to remain anonymous.

Our decision has been reached after consultation between the reviewers. Based on these discussions and the individual reviews below, we regret to inform you that we cannot accept your present manuscript for publication in *eLife*. We felt that you have undertaken a very important subject matter and there are many new and potentially significant advances in the study. However, all three reviewers expressed significant technical concerns regarding the quality of the material pre and post amplification, characteristics and quality of the natural fluid obtained from pigs, and data analysis. They also felt that the conclusions have not been justified by the data. We hope you will find the comments helpful as you revise your work. Our usual policy is that we reject papers unless technical concerns could be addressed within 1-2 months. We feel that the amount of revision work necessary for your study to be competitive would require more work than one could reasonably do in that time frame. If you do gather additional evidence down the road that could satisfy in full the reviewer concerns, we could potentially welcome the submission of a new manuscript for consideration.

*Reviewer #1:*

The authors have undertaken a very important and difficult study. They are correct in emphasizing the need for data of this type. An enormous amount of work went into carrying out this work. There are some very impressive results, but in some areas, the authors may have attempted some interpretations that are beyond what the data can support.

Figure 1 is instructive and essential, although the legend should be expanded. For all of the figures, there are items that are not explained in the legend. Perhaps these are intuitive for the authors. What is the meaning of the red boxes toward the top of the figure? There are abbreviations that are not explained such as OF-EL and UF-EL. Abbreviations are frequently not explained in the text. Some of the font is illegible.

The data in Table 1, Table 2 and Table 3 are quite useful. Particularly the fact that some blastocysts achieve hatching for the Natur-IVF is impressive. Perhaps it would be helpful to show some images of hatching blastocysts. The data in Figure 2–Figure 5 are legitimate and can be made available as supplements, but interpretation now becomes quite speculative beyond the overall trend that changes from the in vivo control are greater for the C-IVF than for the Natur-IVF. The amount of speculation about the possible significance of specific gene changes should be restricted.

The tables and figures related to DMRs are much weaker. All of the qualifications and problems described in the subsection “Three imprinted genes were differentially methylated in C-IVF, but not in Natur-IVF blastocysts, compared to in vivo blastocysts” raise questions about whether it is feasible to interpret the data in the way that is presented. Given concerns about gaps in the pig genome, manual inspection of all genome regions, and failure to detect prominent DMRs such as SNRPN, which is widely conserved across species, raise questions as to whether these analyses and the associated interpretations are meaningful. The legend for Figure 7 is particularly lacking. There is no explanation of the color significance of the blue and red read data. Is red methylated and blue unmethylated?

An attempt to reduce the manuscript to those data and interpretations that are most solid would be useful.

Philosophically, this reviewer is not convinced that use of human reproductive fluids is the way of the future. The analogy to breast milk is valid in some ways but in most ways not. Reproductive fluids are overwhelmingly less accessible than breast milk, and are likely to vary from patient to patient especially in pathologic circumstances. Even for breast milk, which is far more accessible, babies whose mothers are unable to breastfeed do not usually get fed breast milk from other women, but rather receive artificial formulas of greater and greater sophistication. Wet nursing is unlikely to make a big comeback in the future. Rather proteomic analysis of various fluids should make it possible to prepare reproducibly better and better artificial reproductive fluids.

*Reviewer #2:*

The work by Canovas et al. titled: "Epigenetic and gene expression changes derived from assisted reproduction can be mitigated by reproductive secretions" explores the developmental, transcriptomic (RNA seq) and epigenetics (WGBS by using post-bisulfite adapter tagging- PBAT) effects of IVF. The authors tested single pig blastocysts generated by IVF using conventional media (C-IVF) or media with the addition of physiologic secretions obtained from the fallopian tubes (late follicular phase, early luteal phase) or the uterus (called Nature IVF). The control is provided by in vivo blastocysts flashed out of the uterus. Further they tested performance of a new swim up technique as opposed to the more conventional gradient centrifugation methods to select sperm for IVF.

The authors found that:

1) Swim up technique is superior to standard fertilization, resulting in higher blastocyst rate

Nature IVF compared to C-IVF resulted in

2) higher blastocyst state with faster development (higher number of hatched/ hatching blastocysts

3) higher total cell number, that was similar to in vivo blasts

4) less gene expression changes (789 DEG vs 623)

5) Less DNA methylation changes in CpGs (11% vs 15%) while in vivo embryos had an average of 12%. Further they indicate that more imprinted genes or genes involved in epigenetic controls are altered in C-IVF.

The work has important translational value, given that more than 5 million children have been conceived by IVF and that epigenetic and developmental changes have been postulated in children and described in IVF offspring using animal models. Testing novel fertilization and culture techniques that could decrease the epigenetic and developmental problems in offspring is therefore valuable and important. Further, performing single blastocyst analysis is an important technical achievement.

While the study appears to be well designed, there are few but significant technical concerns.

1) First, it is not described in the paper (there is a reference to a website) the standard operating procedure utilized to obtain the physiologic follicular and uterine fluid from pigs and how the media was stored and controlled for contamination. This is a critical aspect of the paper that is missing. For example: is there evidence that late follicular fluid collected from live pigs is always the same, given that changes in nutrition/ temperature/ wellbeing of the animal could change the composition of the fluid? Further: is the fluid from a single animal or pooled from how many animals?

2) Quality controls and validation of RNA seq and WGBS are missing. This is critical given that authors worked with extremely low amount of material. (How was the quality of RNA or RNA amplification assessed? How many reads were obtained per sample? Further, validation of some key genes in independent samples, by RT PCR and bisulfite sequencing is missing).

3) Unsupervised clustering of the 9 samples for both RNA seq and WGBS of all genes (not only the statistically different genes) should be provided to truly confirm that samples cluster based on fertilization and culture methods.

*Reviewer #3:*

In the manuscript entitled "Epigenetic and gene expression changes derived from assisted reproduction can be mitigated by reproductive secretions", Canovas and colleagues investigate the transcriptome and methylome of in vivo-derived and in vitro produced pig blastocysts. To do this, the group first developed a swim-up method for isolating sperm. Following fertilization, in vitro-produced embryo was cultured in conventional conditions that include BSA, or conventional culture augmented with 1% oviductal fluid and uterine fluid. RNA seq and whole genome bisulfite seq were performed on three individual blastocysts per treatment group then data were pooled for analysis. Differential expressed genes and differentially methylated regions were identified between the various experimental groups. In this manuscript, the authors have produced a vast amount of data using novel approaches, including the production of pig embryos with reproductive secretions and single blastocyst methylation analyses. Overall, the paper is well written.

1) Overinterpretation/misrepresentation of data.

A) The manuscript concluded that "reproductive fluids improve the outcome of IVF and the quality of pig blastocysts produced in vitro", and that "remarkable findings were that Natur-IVF blastocysts attained a more advanced developmental stage". These conclusions are not readily discernible from the data. In Table 2 and Table 3, C-IVF embryos produced a great percent of cleavage stage embryos and early blastocysts than Natur-IVF. Additionally, there was no significant difference between C-IVF and Natur-IVF in percent of blastocysts and expanded blastocyst formed or the yield. How does this show an improved outcome or advanced developmental stage? While there was a significant difference in hatching rate, it would appear that embryos were not left long enough to truly assess this, since 0 out of 903 C-IVF embryos and 48 out of 961 (5%) Natur-IVF had hatched.

B) The authors bias the presentation of the transcriptome and methylome data towards finding less difference in the Natur-IVF group. The major questions that should be asked are whether there are expression/methylation changes between the three experimental groups and then where do these differences exist. Furthermore, what is the variation between embryos in the same group? The authors should present PCA graphs to visualize the three embryos and three groups. This should then be followed by pairwise comparisons for all groups, with VENN diagrams shown for all comparisons, including Natur-IVF and C-IVF. In fact, when the comparison is made (subsection “Natur-IVF blastocysts show fewer aberrantly expressed genes than C-IVF blastocysts”, second paragraph), only 29 differentially expressed genes were identified between Natur-IVF and C-IVF, demonstrating little difference in the cultured groups. Instead, as seen in Figure 1, greater differences were found between the in vivo group and the two cultured groups. For DNA methylation, there does not appear to be a significant difference between the three groups for total methylation levels or methylation levels over specific genomic features (Table 4). Here again, C-IVF and Natur-IVF groups need to be compared to determine how much they differ from each other. Finally, while comparisons between the in vivo and each culture group produced a differ subset of genes affected, there does not appear to great differences in the percent of genes misregulated (gene expression: C-IVF 68% up-, 32% down-regulated; Natur-IVF 69% up- and 31% down-regulated; DNA methylation: C-IVF 57% more and 43% less methylated; Natur-IVF 66% more and 34% less methylated). Thus, from the present analysis, it does not appear that "Epigenetic and gene expression changes derived from assisted reproduction can be mitigated by reproductive secretions"(title).

2) In this manuscript, the authors produced single blastocyst profiling then pooled the data. This suggests that only genes affected in all three embryos would be analyzed. Previous DNA methylation analyses have shown that losses and gains of DNA methylation appear to occur stochastically. Thus, pooling of the data may mask gene differences in response to culture. Is there enough coverage to perform analyses on single embryos? If not, can differences be assessed with 2/3 embryos showing the same differences? If not, the authors need to acknowledge that they have analyzed only the most common changes.

3) The authors performed both transcriptome and methyome analyzes. Have they mapped the two data sets to determine whether DNA methylation changes are correlated with gene expression changed in neighboring genes?

[Editors’ note: what now follows is the decision letter after the authors submitted for further consideration.]

Thank you for submitting your article "DNA methylation and gene expression changes derived from assisted reproduction can be decreased by reproductive fluids" for consideration by *eLife*. Your article has been reviewed by three peer reviewers, including Mellissa Mann (Reviewer #3), and the evaluation has been overseen Jessica Tyler as the Senior Editor and Reviewing Editor.

The reviewers have discussed the reviews with one another and the Reviewing Editor has drafted this decision to help you prepare a revised submission.

In summary we feel that this is a valuable contribution to the field. There are numerous concerns but they can be addressed by changes to the text. One of the key concerns that needs to be addressed is the nature of the natural fluid used in the study, because if the same lot were used for all the experiments, this would be a problem. There are also a lot of major grammatical and editorial improvements that are essential, and this work must be edited by a scientist who speaks English as a first language before being sent back to us. I have included the full reviews here, so that you can make all of the necessary changes, as we consider them all essential and very easy to incorporate.

*Reviewer #1:*

The resubmission of the work by Canovas et al. has responded to the majority of comments. However there still remain several concerns. Overall the style of the new version is more difficult to read and will benefit from significant editing both from a language and spelling point of view.

1) In particular the standard operating procedure utilized to obtain the physiologic follicular and uterine fluid from pigs and how the media was stored and controlled for contamination needs to be described in Materials and methods. Referring to other manuscripts is not helpful for readers. For example, the Carrasco 2008 manuscript is not even mentioned in the manuscript. The Coy 2008 paper includes only a short Methods section with a short description of the methods to collect samples from pigs, with no indication on how many pigs were pooled or the range of their age. As an example of lack of clarity, based on the Coy paper: for how many hours are the fallopian tubes at room temperature from sacrificing the animal to collecting the fluid? This is critical to the reader, as quality of the fluid could be compromised or lost during manipulation/transport at room temperature. More Details, including some included in the rebuttal to authors need to be included in Materials and methods section. Given the novelty of the method, this is absolutely needed.

2) The fact that adding only 1% of oviduct fluid can obtain such remarkable results is impressive and this finding is surprisingly missing in the Discussion. If the authors were to add 5% or 10% of OF, would the result change? How did the authors come to the 1% dose? This needs to be commented on in the Discussion.

3) A statement in the Discussion that validation of the RNA and DNA methylation results in independent samples was not performed need to be added as an important limitation of the paper.

4) Please confirm and describe in the Methods section that the 3 biological replicates in the nature IVF group were generated from the addition of 3 independent sets of uterine and oviductal fluids (POF-LF, OF-EL; UF-EL), collected from different animals. This is a key experimental factor that is not described in the Methods section. In fact, adding the same OF to 3 different blastocysts would be expected to lead to similar/ consistent transcriptomic and epigenetic results in blastocysts. If this was not done, data from independent fluid donors should be provided.

*Reviewer #2:*

It was not possible to review the manuscript in much depth as was the case for the original submission. However, in general, the authors appear to have responded well to lengthy criticisms of three reviewers. The interpretations are more restricted. This reviewer continues to doubt that the future of the field of human ART lies in collecting biological fluids from humans or animals but rather that artificial fluids mimicking natural fluids more and more closely are the answer. However, the authors are entitled to their opinion. In general, the manuscript is an excellent contribution to the knowledge in this field.

*Reviewer #3:*

In the manuscript, Canovas and colleagues have investigated an important question in the ART field, for which they produced an impressive amount of data and novel findings. They have clarified and addressed all my concerns.

---

## [Author Response]

[Editors’ note: the author responses to the first round of peer review follow.]

*[…]Reviewer #1:*

*The authors have undertaken a very important and difficult study. They are correct in emphasizing the need for data of this type. An enormous amount of work went into carrying out this work. There are some very impressive results, but in some areas, the authors may have attempted some interpretations that are beyond what the data can support.*

We acknowledge the encouraging comments from the reviewer and are glad to see he/she agrees with the need of this type of study. We have modified the manuscript according to his/her suggestions and those from the other reviewers and think that interpretations are now more adjusted to the data presented. So, we hope this version of our manuscript can be now considered acceptable for publication.

*Figure 1 is instructive and essential, although the legend should be expanded. For all of the figures, there are items that are not explained in the legend. Perhaps these are intuitive for the authors. What is the meaning of the red boxes toward the top of the figure? There are abbreviations that are not explained such as OF-EL and UF-EL. Abbreviations are frequently not explained in the text. Some of the font is illegible.*

Figure 1 and the others have been modified according to the reviewer´s comments. Legends have been expanded, abbreviations have been included and font is now higher. Also, it has been split into Figure 1 and Figure 2 for better clarification.

*The data in Table 1, Table 2 and Table 3 are quite useful. Particularly the fact that some blastocysts achieve hatching for the Natur-IVF is impressive. Perhaps it would be helpful to show some images of hatching blastocysts.*

Thank you for the comment. We have included some images of blastocysts in Table 2.

*The data in Figure 2–Figure 5 are legitimate and can be made available as supplements, but interpretation now becomes quite speculative beyond the overall trend that changes from the in vivo control are greater for the C-IVF than for the Natur-IVF. The amount of speculation about the possible significance of specific gene changes should be restricted.*

The amount of speculation has been restricted. In addition, a PCA analysis has been included showing how the three individuals of each group cluster together, thus legitimising the three pairwise comparisons made for exploring the gene expression data. Also, some figures have been modified and moved to Supplementary files.

*The tables and figures related to DMRs are much weaker. All of the qualifications and problems described in the subsection “Three imprinted genes were differentially methylated in C-IVF, but not in Natur-IVF blastocysts, compared to in vivo blastocysts” raise questions about whether it is feasible to interpret the data in the way that is presented. Given concerns about gaps in the pig genome, manual inspection of all genome regions, and failure to detect prominent DMRs such as SNRPN, which is widely conserved across species, raise questions as to whether these analyses and the associated interpretations are meaningful.*

The qualifications described in relation to the imprinted gene analysis are a legitimateand candid account of how we went about trying to identify candidate imprinted genes in the pig genome, based on the more extensive knowledge of imprinting in mouse and human, and in the face of some deficiencies in the pig genome assembly and the generally poor level of gene annotation. It is recognised that not all imprinted genes are conserved in their imprinted status between species and, indeed, our analysis of the genome organisation of some of these genessuggested that they are unlikely to be imprinted in pig. In the case of SNPRN, in human and mouse this is a very complex imprinted locus in which there are multiple alternative promoters and associated CpG islands that have arisen through a complex series of genomic duplication events (for example, see Smith et al. 2011PLoSGenet. 7:e1002422) so, in the absence of further evidence, we did not feel it justified to attribute to any of possible CpG islands in the pig SNRPN locus the candidate igDMR. The fact that some imprinted genes could not be confidently identified based on comparative sequence organisation does not detract from the fact that we were able to identify 14 candidate igDMRs, which corresponds to more than half of the corresponding regions known from the mouse, and represents a very substantial subset and legitimises the analysis. Amongst these, we did detect significant differences in methylation at 3 igDMRs, and this finding is detailed in the text.

Nonetheless, we would like to thank reviewer 1 for his/her comment because, while preparingthis answer, we have detected some mistakes in the writing that could have been confusing for the reader. We have re-written these paragraphs and hope now the methylation data and the associated interpretations in the manuscript are meaningful.

*The legend for Figure 7 is particularly lacking. There is no explanation of the color significance of the blue and red read data. Is red methylated and blue unmethylated?*

*An attempt to reduce the manuscript to those data and interpretations that are most solid would be useful.*

Legend for Figure 7 has been expanded and corrected because we found some mistakes. We thank the reviewer for this. Red represents methylated cytosines and blue unmethylated. Among the numbers of DMRs found in the unbiased analysis, we identified one overlapping IGF2R in C- IVF embryos compared to in vivo or to Natur-IVF. Also, when the analysis was targeted to CpG islands, we identified one differentially methylated CpGisland in the IGF2R gene. For this reason, and due to the importance of IGF2R in the anomalies observed after ART in animals, we thought it was important and legitimate to include it in this figure as an example of consistent methylation difference in our groups. Similarly, we thought it important to present the result for another igDMR (NNAT) that showed methylation differences between groups, given the importance of correct maintenance of methylation at imprinted genes.

*Philosophically, this reviewer is not convinced that use of human reproductive fluids is the way of the future. The analogy to breast milk is valid in some ways but in most ways not. Reproductive fluids are overwhelmingly less accessible than breast milk, and are likely to vary from patient to patient especially in pathologic circumstances. Even for breast milk, which is far more accessible, babies whose mothers are unable to breastfeed do not usually get fed breast milk from other women, but rather receive artificial formulas of greater and greater sophistication. Wet nursing is unlikely to make a big comeback in the future. Rather proteomic analysis of various fluids should make it possible to prepare reproducibly better and better artificial reproductive fluids.*

We acknowledge this comment and agree partially. Of course, human reproductive fluids are not easy to collect, although we have developed a system to do it and are now getting samples of around 50-150 μL per woman. The trick is to take the samples from healthy ovum donors attending the fertility centers. With the appropriate informed consents and legal permits, we are collecting these “health” samples simply by introducing a special catheter throughthe vagina to the uterus in these donors and characterizing the proteomic profiles as well as other parameters. Bearing in mind that the amount of fluids needed is very small (we have used in the pig model 1% in the final volume of 500 μL) and that the volumes of culture medium in the human clinics are around 50 μL per embryo (or smaller microdrops), each sample collected from only one woman could be potentially used as additive for around 200-400 drops of 50 μL. So, if the beneficial effect were proved, it would not be so difficult to replace the current rHBSA used as protein additive in the human culture media by the corresponding reproductive fluid (previously tested, characterized and endotoxin-free).

Regarding human milk, to use breastfeeding or artificial formula is a personal decision, butmedical benefits of the first one are very clear: “it contains nutrients necessary for infant's growth but also numerous bioactive factors contributing to beneficial effects on gastrointestinal maturation, host defence, infection, cardiovascular risks, metabolic disease, neurodevelopmental outcome as well as in infant's and mother's psychological well-being” (reviewed recently by de Halleux V et al., Semin Fetal neonatal Med 2016). Human milk is the gold standard to provide nutritional support for all healthy and sick newborn infants including the very low birth weight (VLBW) infant (<1500 g) (Johnston M et al. Pediatrics 2012) When own mother's milk it is unavailable, donor human milk is recommended as the first alternative, which could be obtained through human milk biobanks that are growing worldwide (Updegrove KH. Breastfeed Med 2013). Similarly, the idea of creating reproductive fluids biobanks with fully characterized samples could provide some benefits and may be important in the future for treatment of specific pathologies.

*Reviewer #2:*

[…] While the study appears to be well designed, there are few but significant technical concerns.

*1) First, it is not described in the paper (there is a reference to a website) the standard operating procedure utilized to obtain the physiologic follicular and uterine fluid from pigs and how the media was stored and controlled for contamination. This is a critical aspect of the paper that is missing. For example: is there evidence that late follicular fluid collected from live pigs is always the same, given that changes in nutrition/ temperature/ wellbeing of the animal could change the composition of the fluid? Further: is the fluid from a single animal or pooled from how many animals?*

All the samples were pooled from commercial animals raised for human consumption (Large White x Landrace hybrids) from collaborative farms, fulfilling all the requirements for wellbeing, health and nutrition. The animals were slaughtered in an abattoir near the University of Murcia belonging to a big food industry (El Pozo, S.A). We have not included more details because all the procedures for fluid collection and storing were first described in 2008 (Carrasco et al.) and later on we have published different articles using these fluids and testing their effect or the effect of specific oviductal proteins on IVF, zona pellucida hardening, embryo development, embryo survival after cryopreservation, etc. These are the references published until now:

Algarra B, Han L, Soriano-Úbeda C, Avilés M, Coy P, Jovine L, Jiménez-Movilla M. The C-Terminal Region Of Ovgp1 Remodelsthe Zona Pellucida1 And Modifies fertility parameters. Scientific Reports 2016. | 6:32556 | Doi: 10.1038/Srep32556

Lopera-Vásquez R, Hamdi M, Maillo V, Lloreda V, Coy P, Gutiérrez-Adán A, Bermejo-Álvarez P, Rizos D..Effect Of Bovine Oviductal Fluid On Development And Quality Of Bovine Embryos in Vitro. Reproduction Fertility And Development 2015.Doi: 10.1071/Rd15238.

Coy P, Yanagimachi R. Common And Species-Specific Roles Of Oviductal Proteins In Mammalian Fertilization And Embryo Development. Bioscience65:973-984.2015. (Doi:10.1093/Biosci/Biv119).

Ballester L, Romero-Aguirregomezcorta J, Soriano-Úbeda C, Matás C, Romar R, Coy P. Timing Of Oviductal Fluid Collection, Steroid Concentrations And Sperm Preservation Method Affect in Vitro Fertilization Efficiency. Fertility And Sterility. 102:1762-1768. 2014 Http://Dx.Doi.Org/10.1016/J.Fertnstert.2014.08.009..

Mondéjar I, Martínez I,Avilés M, Coy P. Identification Of Potential Oviductal Factors Responsible Of The Zona Pellucida Hardening And Monospermy During Fertilization In Mammals. Biology Of Reproduction 89 (3): 67, 1-8. 2013.

Mondéjar I, Avilés M, Coy P. The Human Is An Exception To The Evolutionarily-Conserved Phenomenon Of Pre-Fertilization Zonapellucida Resistance To Proteolysis Induced By Oviductal Fluid. Human Reproduction28:718-728. 2013.

Grullón La, Gadea J, Mondéjar I, Matás C, Romar R, Coy P. How Is Plasminogen/Plasmin System Contributing To Regulate Sperm Entry Into The Oocyte? Reproductive Sciences 20:1075-1082. 2013doi:10.1177/1933719112473657.

Cebrián-Serrano A, Salvador I, Garcia-Roselló E, Pericuesta E, Pérez-Cerezales S, Gutierrez- Adán A, Coy P, Silvestre Ma. Influence Of Oviductal Fluid On in Vitro Fertilisation, Development And Gene Expression Of in Vitro Produced Bovine Blastocysts. Reproduction In Domestic Animals 48(2): 331-338. 2013

Coy P, García-Vázquez Fa, Visconti P, Avilés M. Roles Of The Oviduct In Mammalian Fertilization. Reproduction144 649–660. 2012.

Coy P, Jiménez-Movilla M, García-Vázquez Fa, Mondéjar I, Grullón L, Romar R. Oocytes Use Plasminogen- Plasmin System To Remove Supernumerary Spermatozoa. Human Reproduction 27(7):1985-1993. 2012.

Mondéjar I, Acuña S, Izquierdo Rico Mj, Coy P, Avilés M. The Oviduct: Functional Genomic And Proteomic Approach. Reproduction In Domestic Animals 47(3): 22-29.2012

Mondéjar I, Grullón La, García-Vázquez Fa, Romar R, Coy P. Fertilization Outcome Could Be Regulated By Binding Of Oviductal Plasminogen To Oocytes And By Releasing Of Plasminogen-Activators During Interplay Between Gametes. Fertility And Sterility2012 97(2):453-461.

Avilés M, Gutiérrez-Adan A, Coy P. Oviductal Secretions: Will They Be Key Factors For The Future Arts?.Molecular Human Reproduction. 2010 16 (12): 896-906.

Coy P, Avilés M. What Controls Polyspermy In Mammals, The Oviduct Or The Oocyte? Biological Reviews Cambphilos Soc. 2010;85(3):593-605.

Coy P, Lloyd R, Romar R, Satake N, Matas C, Gadea J, Holt Wv. Effects Of Porcine Pre-Ovulatory Oviductal Fluid On Boar Sperm Function. Theriogenology74(4):632-642. 2010.

Lloyd R, Romar R, Matas C, Gutiérrez-Adán A, Holt Wv, Coy P. Effects Of Oviductal Fluid On The Generation, Quality And Gene Expression Of Porcine Blastocyst Produced in Vitro.Reproduction 137:679-687. 2009.

Coy P, Cánovas S, Romar R, Saavedra Md, Grullón L, Mondéjar I, Matás C, Avilés M. Oviduct-Specific Glycoprotein And Heparin Modulate Sperm-Zona Pellucida Interaction During Mammalian Fertilization. Proc Nat Acadsci Usa (Pnas), 105:15809–15814. 2008.

Carrasco Lc, Coy P, Avilés M, Gadea J, Romar R.Glycosidase Determination In Bovine Oviduct Fluid At Follicular And Luteal Phases Of The Estrous Cycle. Reproduction, Fertility And Development20 1-10. 2008

Carrasco Lc, Romar R, Avilés M, Gadea J, Coy P. Determination Of Glycosidase Activity In Porcine Oviduct Fluid At The Different Phases Of The Estrous Cycle. Reproduction136: 833–842. 2008.

In the present study, apart from the website mentioned, we have cited in the Methods section our first article describing the beneficial effects of oviductal fluid on the IVF results in pig and cow and the procedures for collection of fluids (Coy and Avilés, 2010, Coy et al., PNAS 2008), as well as two reviews (Coy and Avilés, 2010 and Coy and Yanagimachi, 2015) that summarize the information from the other articles and the procedures for the collection, standardization and quality control of the samples. However, parts of the procedures are patented by University of Murcia (ES-2532659B2) and for this reason we cannot give more details in the paper. Because we are directly involved in Embryocloud, we know that the biological activity of all the samples of fluids used for research purposes, as it was the case, was tested by their ability to induce zona pellucida resistance to proteolytic digestion (see Coy et al., 2008). We have included part of this information in Materials and methods and could include the one that this reviewer considers necessary to avoid any technical concern.

*2) Quality controls and validation of RNA seq and WGBS are missing. This is critical given that authors worked with extremely low amount of material. (How was the quality of RNA or RNA amplification assessed?*

RNA quality was assayed by Bioanalyzer and even though each sample came from asingle blastocyst, RIN score was acceptable, with values between 6.1-8.2. According with a recent report (Sigurgeirsson et al., 2014), “there does not seem to be any justification to set a threshold at any specific RIN but rather it is important to be aware of the effects of low RIN and all samples should preferably be in close range in terms of quality.” In fact, this report showed the biggest effect in differential expression by comparing samples with RIN 10 vs RIN8 (affecting 36% of all the expressed genes), while only 1% genes showed as differentially expressed when comparison was between samples with RIN 8 vs RIN6.

In addition, quality control plots were obtained from SeqMonk, which indicated similar output inall samples and low presence of ribosomal RNA (Figure 9). Although high proportion of ribosomal RNA (rRNA) compared to mRNA represents a typical limitation for RNA-seq studies, the use in different species of Ovation RNA-Seqamplification system(Nugen), which was used for this experiment, have resulted in the low percentages of rRNA fragments (Tariq et al. NucleicAcids Res. 2011; Chitwood et al. BMC Genomics 2013).

Author response image 1.**DOI:**
http://dx.doi.org/10.7554/eLife.23670.023

(Where C means C-IVF embryos, N means Natur-IVF embryos and IV means in vivo embryos).

*How many reads were obtained per sample?*

These are the numbers for RNASeq and PBAT,respectively:

ID sample*RNA*-*seq*Mapped READSC937155950C946765974C966781941N276008873N544615700N604043748IV1896431837IV1906750663IV1916278365

ID SamplePBATUnique alignmentsC3413,150,508 (59.2%)C3542,208,651 (54.6%)C3634,002,302 (58.9%)N8533,153,208 (58.5%)N8640,676,107 (52.0%)N5528,310,723 (54.5%)IV18635,990,635 (53.3%)IV19731,379,752 (56.9%)IV19332,812,689 (53.5%)

Regarding the quality of the PBAT data, we are using a low-cell protocol that is now wellestablished in our lab and was optimised for the analysis of a similar number of oocytes (~100) as the number of cells in the individual pig blastocysts (see Stewart et al. 2015 Genes Dev.29:2449) and which has proven to be very robust (the highly stereotypical oocyte methylation landscape provides an excellent reference for the reproducibility of the low-cell PBAT method). The pig blastocyst PBAT libraries had slightly higher sequence duplication rates and lower overall complexity, which probably reflects a slightly lower number of cells the single embryos. The unique sequence alignment rate (table above) is somewhat lower than we typically obtain by mapping PBAT data from mouse or human samples (65-70%), and this likely reflects the gaps in the pig genome assembly.

Further, validation of some key genes in independent samples, by RT PCR and bisulfite sequencing is missing).

We had considered performing qPCR studies to ‘re-validate’ some of our gene- expression findings but there is little evidence that qPCR analyses from the same samples will add any extra utility to our data. Previous studies have shown extremely close correlations between qPCR and RNAseq data (Griffith M et al. (2010) Nat Methods 7: 843–847, Asmann YW, et al. (2009). BMC Genomics 10: 531, Wu AR, et al. (2014). Nat Methods 11: 41–46, Shi Y and He M (2014). Gene 538: 313–322). In contrast to microarrays, which always require qPCR validation, in RNAseq experiments probe bias, poor sensitivity and reduced linear range are not as problematic, since the entire transcript is assessed in a more or less unbiased manner (Wang Z, Gerstein M, Snyder M (2009) Nat Rev Genet 10: 57–63). In cases where there are discrepancies between RNAseq and qPCR, it would be most likely due to bias in the qPCR experiment (which has its own probe-bias based on what region of the cDNA is amplified). Ideally, we could re- validate our findings (potentially by qPCR) in a separate cohort of samples collected for a new experiment including new animals for the in vivo embryos (without guarantee of similarity with the ones already used), but we would consider to perform it only if reviewers consider it critical to publish our results.

The same reason would apply for PBAT DNA methylation analysis by sequencing.

*3) Unsupervised clustering of the 9 samples for both RNA seq and WGBS of all genes (not only the statistically different genes) should be provided to truly confirm that samples cluster based on fertilization and culture methods.*

For RNASeq, the PCA analysis gave the plot which has been included in the manuscript (Figure 3).

For DNA methylation, we got the PCA plot which explains (32%) how in vivo and N-IVF are close each other and also shows the difference among the three C-IVF embryos (Figure 5).

*Reviewer #3:*

[…] 1) Overinterpretation/misrepresentation of data.

*A) The manuscript concluded that "reproductive fluids improve the outcome of IVF and the quality of pig blastocysts produced in vitro", and that "remarkable findings were that Natur-IVF blastocysts attained a more advanced developmental stage". These conclusions are not readily discernible from the data. In Table 2 and Table 3-IVF embryos produced a great percent of cleavage stage embryos and early blastocysts than Natur-IVF. Additionally, there was no significant difference between C-IVF and Natur-IVF in percent of blastocysts and expanded blastocyst formed or the yield. How does this show an improved outcome or advanced developmental stage? While there was a significant difference in hatching rate, it would appear that embryos were not left long enough to truly assess this, since 0 out of 903 C-IVF embryos and 48 out of 961 (5%) Natur-IVF had hatched.*

We have modified the tables to make them clearer, because from the reviewer´scomments we extracted that we were not able to explain it properly. In Table 2, C-IVF embryos cleaved 5% more than Natur-IVF at day 2 (47.5 ± 1.6 vs 42.1 ± 1.6). However, this difference was not maintained through the development since the proportion of blastocysts in both groups was similar on day 7.5 (41.4 ± 2.4 vs 44.5 ± 2.5). Nonetheless, both percentages were higher than the best results previously described (Redel et al., 2016. Mol Reprod Dev), and it could be explained because oocytes in the C-IVF group also were incubated with oviductal fluid before IVF, as shown in the Figure 2. In addition, Natur-IVF group (whose embryos were in addition exposed to reproductive fluids during embryo culture) showed higher number of cell per blastocysts than C- IVF, (49.9 ± 3.7 vs 81.8 ± 7.2) and it was similar to in vivo embryos (87.0 ± 7.2). With these arguments we think that we can state that reproductive fluids improve the outcome of IVF and the quality of pig blastocysts produced in vitro.

In Table 3 (now Table 2.B) the higher percentage of early stage blastocysts in C-IVF group (57 out of 178) is due to the delay in their developmental kineticscompared to Natur-IVF group. The different blastocyst stages assessed in Table 3 (now Table 2.B) correspond with the total of blastocysts produced with each system (table 2.A) and, on day 7.5, only 12.8% of them (23 out of 180) were early blastocysts in the Natur-ART group (compared to 31.7% in C-IVF, 57 out of 178) because most of them had progressed to more advanced stages, such as blastocyst (55 embryos) expanded (65 embryos), hatching (28 embryos) or hatched (9 embryos). We thank you for thiscomment and have modified this Table adding information in order to clarify the data provided.

*B) The authors bias the presentation of the transcriptome and methylome data towards finding less difference in the Natur-IVF group. The major questions that should be asked are whether there are expression/methylation changes between the three experimental groups and then where do these differences exist. Furthermore, what is the variation between embryos in the same group? The authors should present PCA graphs to visualize the three embryos and three groups. This should then be followed by pairwise comparisons for all groups, with VENN diagrams shown for all comparisons, including Natur-IVF and C-IVF. In fact, when the comparison is made (subsection “Natur-IVF blastocysts show fewer aberrantly expressed genes than C-IVF blastocysts”, second paragraph), only 29 differentially expressed genes were identified between Natur-IVF and C-IVF, demonstrating little difference in the cultured groups. Instead, as seen in Figure 1, greater differences were found between the in vivo group and the two cultured groups.*

Thank you for this comment. We have included PCA graphs to show how the three embryos from each group cluster together. As reviewer points, "the major questions that should be asked are whether there are expression/methylation changes between the three experimental groups and then where do these differences exist". That is exactly what we tried to do here. After checking the quality of the data and the clustering of the embryos (now we have added the PCA plot), we did pairwise comparisons and found that, for gene expression analysis, and despite of individual variability, both in vitro groups were more similar between them than any of them compared to in vivo. For this reason, we started analyzing differences between each in vitro group compared to the in vivo one (789 and 623 genes out of 19,638) but continued looking at differences between the two in vitro groups (29 genes out of 19,638). First conclusion was that all the groups were similar, having only 4%, 3.17% and 0.15% DEGs, respectively, for each pairwise comparison. However, we also found that the general tendency for the Natur-IVF DEGs was to show intermediate values between the in vivo and C-IVF ones. And, finally, despite the scarce differences, we found that some DEGs between C-IVF and Natur-IVF could be critical because the corresponding knock out or knock-down studies in mice showed phenotypes of altered/abnormal growth/size, reproduction/fertility, mortality/aging, hematopoietic system, homeostasis/metabolism and other abnormalities. Bearing in mind that the alteration of just one gene could have consequences in the normal development of an embryo, we thought it was important to highlight some of the most representative genes. Altogether, we think that our data "confirm that in vitro culture significantly alters embryonic gene expression”.

*For DNA methylation, there does not appear to be a significant difference between the three groups for total methylation levels or methylation levels over specific genomic features (Table 4). Here again, C-IVF and Natur-IVF groups need to be compared to determine how much they differ from each other. Finally, while comparisons between the in vivo and each culture group produced a differ subset of genes affected, there does not appear to great differences in the percent of genes misregulated (gene expression: C-IVF 68% up-, 32% down-regulated; Natur-IVF 69% up- and 31% down-regulated; DNA methylation: C-IVF 57% more and 43% less methylated; Natur-IVF 66% more and 34% less methylated). Thus, from the present analysis, it does not appear that "Epigenetic and gene expression changes derived from assisted reproduction can be mitigated by reproductive secretions"(title).*

In the context of demethylation that happens during early embryo development, withmean values of DNA methylation around 12% in vivo (12.33 ± 3.6) the observed differences (Natur-IVF: 11.09 ± 2.6 and C-IVF: 15.02 ± 3.3) could have a significant biological effect. In addition, specific DMRs, especially in the context of imprinting regions can also impact embryo development. We never hypothesized to find huge DNA methylation differences, because during the blastocyst stage there is an almost complete resetting of the methylome.

For DNA methylation, the differences were, indeed, subtle, as shown in Table 4, andless than4,000 DMR were found from the total of 258,885 probes created in the unbiased analysis, which represents 1.5%. Here again, C-IVF vs Natur-IVF groups were compared as well as C-IVF vs in vivo and Natur-IVF vs in vivo. We apologise for not having included the comparison between the two in vitro groups in Figure 5. Now it is included, with 3112 DMRs between both groups. Recognising these rare differences, we tried then to search for specific DMRs of obvious potential biological significance, thinking again of the impact that altered regulation of just a single gene could have for correct embryo development. For this reason, we first focused on the specific DMRs characterizing exclusively in each group (Figure 5) and second on the imprinted igDMRs. We were aware that by extracting only the DMRs with a consistent 5% significant change in all the three embryos of one group compared to the other we could be missing other differences occurring only in one individual but we decided this approach as an attempt of collecting at least a first list of potential DMRs having the highest probability to occur. Since we are providing here the first datasets of single pig blastocysts methylome and transcriptome, there are now multiple options of different valid analyses in future studies.

*2) In this manuscript, the authors produced single blastocyst profiling then pooled the data. This suggests that only genes affected in all three embryos would be analyzed. Previous DNA methylation analyses have shown that losses and gains of DNA methylation appear to occur stochastically. Thus, pooling of the data may mask gene differences in response to culture. Is there enough coverage to perform analyses on single embryos? If not, can differences be assessed with 2/3 embryos showing the same differences? If not, the authors need to acknowledge that they have analyzed only the most common changes.*

Thank you for the comment, we agree with you. We have indeed indicated that wechose the option of analyzing only the most common changes in this study, as above explained.

*3) The authors performed both transcriptome and methyome analyzes. Have they mapped the two data sets to determine whether DNA methylation changes are correlated with gene expression changed in neighboring genes?*

Yes, we did it and we did not found significant correlation between both changes.Although DNA methylation at the promoter/gene bodies is directly/indirectly correlated with gene expression, this is not strictly true during the periods of dramatic loss of DNA methylation, as during early embryo development or primordial germ cells (PGC) formation. For example, Gounktela et al., 2015 showed that during demethylation of PGCs there is a general uncoupling between DNA methylation and gene expression at some stages; literally they say: “Our data reveal a remarkable and pervasive loss of DNA methylation in human PGCs and AGCs during prenatal life that has almost no relationship to changes in gene expression”. We did correlation analysis and saw this lack of correlation between our methylation and gene expression data. In our opinion, at this stage of development and with this low level of methylation, this is not a surprising result.

[Editors' note: the author responses to the re-review follow.]

*[…] Reviewer #1:*

*The resubmission of the work by Canova et al. has responded to the majority of comments. However there still remain several concerns. Overall the style of the new version is more difficult to read and will benefit from significant editing both from a language and spelling point of view.*

We acknowledge the encouraging comments from the reviewer and they have been very useful to improve the manuscript. We have modified it according to his/her suggestions and those from the other reviewers. An extensive review of the grammar and spelling has been done by Dr. Gavin Kelsey.

*1) In particular the standard operating procedure utilized to obtain the physiologic follicular and uterine fluid from pigs and how the media was stored and controlled for contamination needs to be described in Materials and methods. Referring to other manuscripts is not helpful for readers. For example, the Carrasco 2008 manuscript is not even mentioned in the manuscript. The Coy 2008 paper includes only a short Methods section with a short description of the methods to collect samples from pigs, with no indication on how many pigs were pooled or the range of their age. As an example of lack of clarity, based on the Coy paper: for how many hours are the fallopian tubes at room temperature from sacrificing the animal to collecting the fluid? This is critical to the reader, as quality of the fluid could be compromised or lost during manipulation/transport at room temperature. More Details, including some included in the rebuttal to authors need to be included in Materials and methods section. Given the novelty of the method, this is absolutely needed.*

We apologize for the lack of clarity about the operating procedure used to obtain the follicular, oviductal and uterine fluids from pigs. We tried to summarize the more relevant information and to avoid an extensive Materials and methods section. However, following reviewer´s comment we have now included detailed information in this section:

“Collection and processing of follicular, oviductal and uterine fluids

Fluids were obtained from animals raised at a commercial farm (CEFU, S.A., Murcia, Spain) and slaughtered in an abattoir belonging to a food industry (El Pozo, S.A) near the University of Murcia. […] One ml follicular fluid (FF) aliquots were stored at -80°C until their use as additives for the IVM medium.”

This is the standard protocol used for porcine IVM in the last 25 years in our laboratory following Naito et al. indications (Naito et al., Gamete Res. 1988 21(3):289-95) and has been proven to provide a beneficial microenvironment for further development of the immature oocytes (e.g., Ito et al., 2008 and Bijttebier et al., 2008).

For the collection of oviductal (OF) and uterine (UF) fluids, genital tracts from cyclic Large White sows (2-4 years old) were obtained at the abattoir and transported to the laboratory on ice within 30 min of slaughter. The cyclic stage of animals was assessed once in the laboratory, on the basis of ovarian morphology on both ovaries from the same female. Oviducts and uteri were classified as early follicular, late follicular, early luteal or late luteal phase (Carrasco et al., 2008). Both oviducts and uteri coming from the same genital tract were classified as in the same stage of the cycle. Once classified, oviducts and uteri were separated and quickly washed once with 0.4% v/v cetrimide solution and twice in saline. Oviducts and uteri were dissected on Petri dishes or trays, respectively, sitting on ice. Once dissected, OF were collected by aspiration with an automatic pipette by introducing a 200µl pipette tip into the ampulla and manually making an increasing pressure gradient from the isthmus to the ampulla. The UF was collected by making a manual increasing pressure gradient from the proximal end to the distal end (utero-tubal junction) of the uterine horn and letting the fluid drop into a sterile 50 ml Falcon tube. Once recovered, samples (OF and UF) were centrifuged twice at 7000 g for 10 min at 4°C to remove cellular debris. Then the supernatant was immediately stored at -80°C until use. Oviducts from animals at the late follicular phase (POF-LF) and at the early luteal phase (POF-EL) gave a mean volume of around 50µl and 40µl, respectively per oviduct. At the early luteal phase, approximately 10 ml of UF per uterine horn were collected each time. Aliquots of 50µl OF and 50ml UF of pooled samples from at least 20 animals for OF and 5 animals for UF were used. Only samples that passed quality controls (pH 7.0-7.6, osmolality 280-320 mOsm/kg, endotoxin <0.10 EU/mL, a minimum 90% of Metaphase II oocytes after IVM with FF and ZP hardening for oocyte preincubation in POF-LF > 1 hour) were used for experiments.

2) The fact that adding only 1% of oviduct fluid can obtain such remarkable results is impressive and this finding is surprisingly missing in the Discussion. If the authors were to add 5% or 10% of OF, would the result change? How did the authors come to the 1% dose? This needs to be commented on in the Discussion.

We acknowledge the encouraging comment from the reviewer.

The use of 1% of fluid was established after testing a large number of experimental conditions in our laboratory where cleavage rate, monospermy, morphology and blastocyst development were assayed (for example, in Ballester et al., 2014 we published the first IVF results adding 1% OF to the IVF medium). Similarly, our collaborator Dr. Rizos, after testing different OF concentrations in the cow model, published that OF concentrations (1.25% and 0.625%) supported embryo development until Day 9 (27.5%) and produced higher-quality blastocysts (Lopera-Vasquez et al., 2015).

We agree that this finding is significant and it has been included in the Discussion.

3) A statement in the Discussion that validation of the RNA and DNA methylation results in independent samples was not performed need to be added as an important limitation of the paper.

Following this request, we have introduced the following paragraph in the Discussion section:

“Considering that previous studies have shown extremely close correlations between qPCR and RNA-seq data (Griffith M et al. (2010) Nat Methods 7: 843–847; Asmann YW et al. (2009) BMC Genomics 10: 531; Wu AR, et al. (2014). Nat Methods 11: 41–46) and that validation by qPCR has its own probe-bias based on what region of the cDNA is amplified, we deem, in contrast to microarrays data, that there is not solid evidence that validation of the RNA-Seq and DNA methylation results by qPCR will provide extra significance to our results. For this reason, we did not perform qPCR validation in this study”.

*4) Please confirm and describe in the Methods section that the 3 biological replicates in the nature IVF group were generated from the addition of 3 independent sets of uterine and oviductal fluids (POF_LF, OF-EL; UF-EL), collected from different animals. This is a key experimental factor that is not described in the Methods section. In fact, adding the same OF to 3 different blastocysts would be expected to lead to similar/ consistent transcriptomic and epigenetic results in blastocysts. If this was not done, data from independent fluid donors should be provided.*

Thank you for the comment. It has helped us to clarify in the manuscript that the observed effect in the Natur-IVF group is not the result of adding reproductive fluid from one specific animal, which would be not representative. As described in the new Materials and methods section reproductive fluids were pooled from different animals (at least 25 females for FF, 20 animals for OF and 5 animals for UF), collected simultaneously to form a large batch/lot of fluid that was homogenous for the different experiments. This experimental design aims to follow the very well established and routine practice for cell cultures or IVF-EC assays that promotes the use of serum from the same lot/batch for a set of experiments. Otherwise, assuming that some variability could exist between animals (although very high similarity exists in age, health status, weight, nutrition, etc.) individual effect and treatment effect could be masked or confused.

We had to use different batches of fluids for the oocyte preincubation step during the in vitro production of embryos because of the volume of fluid needed (see Table 2 where 961 oocytes were used for the Natur-IVF group which means 961 microliters of OF just for the preincubation step). But we used the same batch of fluids for the remaining steps and for the 6 blastocysts used for the DNA and gene expression analysis because to have done otherwise would not have been correct in our opinion. It would have been the same as using different batches of BSA for the C-IVF group or different ages or breeds of animals for the in vivo group. In all the ART laboratories one of the main objectives is the use of standardized protocols and, for example when serum is added, the same batch is used for long periods of time.

We do not consider that data from independent fluid donors can be useful and representative for these experiments. The use of big homogenous batches of follicular fluids is the standard protocol used for porcine IVM in the last 25 years in our laboratory and other groups around the world, following Naito et al. indications (Naito et al., Gamete Res. 1988 21(3):289-95) and it has been demonstrated to provide a beneficial microenvironment for further development of the immature oocytes (e.g., Ito et al., 2008 and Bijttebier et al., 2008). Similarly, the use of large and homogenous batches of OF has been proved to be consistently beneficial by our group when added during IVF (Ballester et al., 2014, Fertility & Sterility 102(6): 1762-1768). Finally, the addition of UF, despite being used in this article for the first time, has been checked in our lab with different batches and always has shown an improvement compared with conventional protocols (e.g. BSA).

*Reviewer #2:*

*It was not possible to review the manuscript in much depth as was the case for the original submission. However, in general, the authors appear to have responded well to lengthy criticisms of three reviewers. The interpretations are more restricted. This reviewer continues to doubt that the future of the field of human ART lies in collecting biological fluids from humans or animals but rather that artificial fluids mimicking natural fluids more and more closely are the answer. However, the authors are entitled to their opinion. In general, the manuscript is an excellent contribution to the knowledge in this field.*

We really appreciate the reviewer´s comments, including the scepticism about the future of the field of human ART. Only the future will reveal the right answer. We are glad to see he/she recognizes our work as “an excellent contribution to the knowledge in this field”.